# Dynamic Resource Allocation for Network Slicing with Multi-Tenants in 5G Two-Tier Networks

**DOI:** 10.3390/s23104698

**Published:** 2023-05-12

**Authors:** Jia-You Lin, Ping-Hung Chou, Ren-Hung Hwang

**Affiliations:** 1Department of Computer Science and Information Engineering, National Chung Cheng University, Chiayi 62102, Taiwan; ljy110p@cs.ccu.edu.tw (J.-Y.L.); ghuiry2546@ccu.edu.tw (P.-H.C.); 2Department of Computer Science, College of AI, National Yang Ming Chiao Tung University, Tainan 71150, Taiwan

**Keywords:** network function virtualization, network slicing, multi-access edge computing, optimal resource allocation, dynamic offloading

## Abstract

Virtualization is a core 5G network technology which helps telecom companies significantly reduce capital expenditure and operating expenses by deploying multiple services on the same hardware infrastructure. However, providing QoS-guaranteed services for multi-tenants poses a significant challenge due to multi-tenant service diversity. Network slicing has been proposed as a means of addressing this problem by isolating computing and communication resources for the different tenants of different services. However, optimizing the allocation of the network and computation resources across multiple network slices is a critical but extremely difficult problem. Accordingly, this study proposes two heuristic algorithms, namely Minimum Cost Resource Allocation (MCRA) and Fast Latency Decrease Resource Allocation (FLDRA), to perform dynamic path routing and resource allocation for multi-tenant network slices in a two-tier architecture. The simulation results show that both algorithms significantly outperform the Upper-tier First with Latency-bounded Overprovisioning Prevention (UFLOP) algorithm proposed in previous work. Furthermore, the MCRA algorithm achieves a higher resource utilization than the FLDRA algorithm.

## 1. Introduction

As Fifth Generation (5G) networks become increasingly deployed [1], demands for new services and multiple-tenant models have increased accordingly. To enable the provision of 5G native features which satisfy the stringent Quality-of-Service (QoS) [2,3] requirements of the network users, cellular operators must upgrade their core networks through the introduction of new equipment or computing paradigms. Unlike Fourth Generation (4G) network models, in which the network equipment is tied to proprietary vendors [4], 5G is built virtually on software-based platforms such as Central Office Re-architected as a Datacenter (CORD) [5], where these platforms leverage Software Defined Network (SDN) [6], Network Function Virtualization (NFV), and cloud computing technologies to build agile datacenters at the network edge. By providing different services on the same commodity hardware infrastructure, network virtualization and software-based networking enable telecom operators to break free from the shackles of private hardware and develop more robust infrastructures for the provision of multiple services.

With the development of virtualization technology, it is now possible to share the communication resources of 5G networks across multiple telecom operators. Thus, the feasibility of developing diversified and convenient services under 5G networks has attracted great attention in recent years. In the context of 5G networks, the concept of 5G Quality Indicator (5QI) [7] has emerged as an important consideration in evaluating the service quality characteristics of the network. The 5QIs are utilized to describe different types of data transmissions, with each type requiring specific 5QIs to ensure effective transmission over the network. As such, the 3rd Generation Partnership Project (3GPP) standards identify three services (slices) to support the QoS requirements of different 5G applications, namely:Enhanced Mobile Broadband (eMBB) [8]: provides enhanced bandwidth transmission speeds of up to 10 Gbps to improve existing communication services and offer users a seamless transmission experience. The slice is currently utilized for applications such as streaming and broadcasting, live video, and Augmented Reality/Virtual Reality (AR/VR).Ultra-Reliable and Low Latency Communications (URLLC) [9]: intended for communication applications with high reliability (i.e., error rates less than $10^{-5}$) and low time latency (i.e., less than 1 ms) requirements. The slice is currently allocated mainly to critical applications such as remote surgery and autonomous driving.Massive Machine Type Communications (mMTC) [10]: provides the means to satisfy the communication needs of up to 1 million connected devices per square kilometer. The slice is currently used mainly to support the communication requirements of Internet of Things (IoT) devices.

In addition, regarding the relevant literature on QoS and Quality of Experience (QoE) in the entire telecommunications system, Poryazov et al. [11] proposed a method for predicting QoE parameters in an overall telecommunications system (both users and the telecommunications network). This method was based on predicting QoS indicator values and outlined four normalization techniques. Additionally, Poryazov et al. proposed a proportional normalization method for indicators and provided numerical illustrations. This approach was suitable for various types of telecommunications networks and could consider QoS and QoE indicators at different levels. Manzanares-Lopez et al. [12] presented a novel approach to managing QoS in IEEE 802.11 wireless networks using a Software-Defined Networking (SDN) framework. The SDN paradigm is traditionally used to configure forwarding elements in wired networks, but the authors proposed an extension of SDN to wireless networks and the use of OpenFlow protocol to configure forwarding elements. In addition, the authors proposed a mathematical model that calculated “gain” to adapt to the network dynamics. The proposed SDN-based framework took advantage of centralized network management to respond to dynamic QoS requirements in 802.11 networks by applying the Dynamic Enhanced Distributed Channel Access (DEDCA) mechanism.

To satisfy the privacy and service quality assurance requirements of the different tenants in 5G networks, the communication and computing resources of each tenant must be independent and exclusive of those belonging to the other tenants in the network. This can be achieved through the network slicing paradigm [13]. For instance, Chen et al. [14] address a network slicing problem that involves mapping multiple customized virtual network requests (also known as services) to a shared network infrastructure and allocating network resources to satisfy diverse QoS requirements. To achieve optimal QoS-aware network slicing in service-oriented networks, they propose a mixed integer nonlinear programming formula that optimizes resource consumption while considering QoS requirements, flow routing, and resource budget constraints. However, the problem of allocating the computing and communication resources of the network across multiple tenants and slices in an optimal manner remains a significant challenge.

The main contributions of this paper are as follows:We proposed a method for dynamic resource allocation and task offloading optimization for 5G two-tier multi-tenant network slicing that considered the real-time resource usage of nodes in the network model. This enabled the determination of optimal routing and resource allocation for new service requests, with the goal of satisfying QoS requirements and maximizing resource utilization. Therefore, our approach effectively avoided problems such as premature resource depletion and blocking due to excessive resource allocation to nodes.Developed Minimum Cost Resource Allocation (MCRA) and Fast Latency Decrease Resource Allocation (FLDRA) dynamic resource allocation algorithms that worked with Cloud-only (*CO*), Edge-only (*EO*), and Parallel Cloud-Edge Hybrid (*PH*) network structures. Simulation results demonstrated that the MCRA algorithm calculated weights based on the current resource usage of each node to achieve optimal resource availability. Secondly, the FLDRA algorithm prioritized minimizing the node with the highest E2E delay while satisfying multi-tenant service quality requirements. Therefore, the effectiveness is also superior to Upper-tier First with Latency-bounded Overprovisioning Prevention (UFLOP). On the other hand, among our proposed three network models, the *CO* model experienced the earliest blocking, followed by the *EO* model, and finally the *PH* model. The reason is that the *PH* model supports offloading to the cloud. When comparing the *CO* and *EO* models, due to the difference in the number of nodes, under the same latency constraints, the *EO* model consumes fewer resources per request. In other words, the *PH* model can accommodate a higher arrival rate based on the same blocking rate. Overall, these algorithms and models significantly reduced service request blocking rates and improved resource utilization.Our proposed resource allocation approach considered the varying QoS requirements of emerging 5G applications for multi-tenant and multi-network slicing. It ensured that each tenant’s service quality assurance requirements were satisfied by optimizing resource allocation for each tenant’s network slice, thereby enhancing the overall efficiency and effectiveness of the multi-tenant 5G network and improving the quality of service provided to tenants.

The remainder of this paper is organized as follows. Section 2 describes previous related work on the network slicing paradigm. Section 3 introduces the two-tier system model considered in the present study and formulates the problem definition. Section 4 introduces the resource allocation flow and latency models adopted in the present analysis. Section 5 describes the method used to allocate the resources in the three network structure models and introduces the two dynamic resource allocation algorithms proposed in this work. Section 6 presents and discusses the simulation results obtained for the average delay, resource availability, and blocking rate under the various resource allocation algorithms and network structure models. Finally, Section 7 provides some brief concluding remarks.

## 2. Related Works

Existing studies on the allocation of computing and communication resources to different network slices in 5G networks can be broadly classified based on their architectures as either one-tier schemes [15,16,17,18,19], two-tier schemes [20,21,22,23,24], as well as Graph-based deep learning and reinforcement learning schemes [25,26,27,28,29]. The related studies are reviewed in the following sub-sections.

### 2.1. One-Tier Architecture

Xu et al. [15] applied the concept of end-to-end (E2E) network slicing to next-generation SDN-based mobile network solutions. However, the problem of sharing the Radio Access Network (RAN) among multiple mobile users was not addressed. Rostami et al. [16] considered the problem of layering across wireless access and transmission networks in 5G Multi-access Edge Computing (MEC) architectures and presented an approach for the design and implementation of programmable orchestration architectures. Katsalis et al. [17] proposed an architecture for supporting full E2E (cloud to RAN) slices in 5G networks. However, the authors did not consider the allocation of the RAN resources across multiple users or the strict latency requirements of certain 5G services, such as URLLC. Li et al. [18] presented a resource management and control strategy for edge and cloud collaborative computing, but provided only a brief discussion of the considered factors and conditions. Wang et al. [19] developed an approach for the effective allocation and adjustment of edge resources by dynamically adjusting the weights of each edge cloud using four different weights. However, the authors analyzed only the edge resources of the architecture, i.e., the network resources were not considered.

In general, the studies in [15,16,17,18,19] limit the network slicing problem to a one-tier architecture with only a single computing resource and ignore the architectures of the cloud and edge networks. Furthermore, even though the study in [19] presents an approach for dynamically adjusting the weights of the edge resource, the dynamic adjustment process takes account of only the edge computing resources and ignores the communication resources.

### 2.2. Two-Tier Architecture

Zhang et al. [21] proposed a 5G network slicing framework for efficient Virtual Network Function (VNF) deployment that included both the Edge Servers (E) and the Cloud Servers (C). However, the authors did not address the problem of satisfying different service requirements when allocating the network resources, instead focusing only on the allocation of the Central Processing Unit (CPU) and memory and the throughput required by the VNFs. Ren et al. [22] proposed a collaboration strategy between the cloud and edge computing tiers of the architecture which allowed the tasks of the mobile devices to be shared between the edge nodes and cloud servers. However, the latency of the transmissions between the edge nodes and the servers was not considered. Wang et al. [23] presented a latency-based workload allocation scheme for minimizing the energy consumption and satisfying the latency requirements in IoT edge-cloud computing systems. However, the study focused mainly on maximizing the tradeoff between the energy consumption and the latency guarantee under fixed resource scenarios rather than providing each user with an appropriate resource allocation while ensuring compliance with the latency constraints. Guo et al. [24] proposed a multi-layered data stream processing system that combined the computing power of the cloud center with the close-range advantage of MEC and aimed to optimize the task allocation problem in such a way as to minimize the latency. However, the proposed approach failed to guarantee the QoS satisfaction of the various slices.

Overall, while the studies in [23,24] propose a comprehensive approach for leveraging the computing and network resources of both the edge devices and the cloud, they fail to consider the delay constraints specified in the Service-Level Agreements (SLAs) of the slices. Chien et al. [20] proposed an E2E network slicing problem using a single edge server and CORD as an example. However, as described more fully in Section 3.1, the method proposed in [20] suffers three important limitations: (1) it does not consider the core network, (2) it does not account for the lack of queueing models, including improper consideration of the tandem queues and the handling of each service request separately, and (3) it does not propose an optimization strategy for service triage.

### 2.3. Graph-Based Deep Learning and Reinforcement Learning

Tam et al. [25] reviewed numerous studies on optimizing control policies using Graph Neural Networks (GNNs), including offloading strategies, routing optimization, virtual network function orchestration, and resource allocation. The authors also discussed the relationship between GNNs and communication networks by exploring the specificity of input network topologies and device/resource abstractions. Moreover, they identified possible input features, processing procedures, and target applications (e.g., congestion prediction, rule configuration) that could collaborate with GNNs in network management and orchestration entities. Jiang [26] focused on exploring the various applications of graph-based deep learning techniques in this domain. GNNs were shown to possess the ability to capture complex relationships between objects and reason about data described in graph form. Moreover, their effectiveness was demonstrated in a variety of prediction tasks at the node, edge, and graph levels. Yuan et al. [27] proposed combining deep reinforcement learning and graph convolutional neural networks method to achieve more intelligent and efficient resource allocation by learning the relationships between different nodes in the wireless network in hybrid overlay-underlay cognitive radio networks. Finally, they discussed its potential applications in future wireless communication systems after validating and analyzing the method through simulation experiments. Shao et al. [28] aimed to address the real-time inter-slice resource management strategy design problem in a dense cellular network scenario. The authors formulated the problem as a multi-agent reinforcement learning problem and leveraged graph attention network to improve resource management efficiency. Experimental results showed that the proposed approach coped with challenges such as frequent base station handover and fluctuations in different service requirements, and achieved good performance. Dong et al. [29] investigated how network slicing technology divides a common infrastructure into multiple logical networks to support services with different requirements in 6G mobile systems. However, as a network management method, it was difficult for network slicing to achieve real-time resource allocation to satisfy the stringent requirements of services in 6G networks. Therefore, the authors proposed a joint network slicing and routing mechanism that combined the network management and control framework to provide fine-grained, real-time, and dynamic resource allocation. Specifically, they proposed a Graph Convolutional Network (GCN)-powered Multi-Task Deep Reinforcement Learning (DRL) model to solve this complex resource allocation problem. The DRL model was first extended into a multi-task mode, where multiple output branches were matched to jointly schedule resources in every network slice. Finally, the effectiveness and superiority of the proposed approach were demonstrated through experiments.

However, despite the remarkable achievements of graph-based deep learning and reinforcement learning schemes in resource allocation and management in communication networks, queueing methods remain a highly useful, classic, and deep approach. Simulating a queueing network can transform resource allocation problems into queueing problems, which facilitates the design of strategies to make allocation policies more intuitive and efficient. Furthermore, queueing methods typically have lower computational costs and do not require large amounts of training data. In summary, while graph-based deep learning and reinforcement learning schemes offer benefits in communication network management and resource allocation, queueing methods also have unique advantages that can be used to better design and implement resource allocation strategies in network scenarios.

Table 1 summarizes the scope and capabilities of the resource slicing solutions proposed for next-generation 5G mobile networks in the studies above. For studies that do not utilize graph-based deep learning and reinforcement learning techniques, the table shows the study’s high-level scope (e.g., concept, model, algorithm) and the functions and entities involved (network/server slicing, RAN, E, Core Network (CN), and C). As in the discussions above, the studies are classified broadly in terms of their architecture, i.e., one-tier or two-tier. Furthermore, for each study, the table indicates whether the resource allocation at different tiers is dynamically adjustable and whether or not the proposed solution satisfies the latency constraints imposed in the tenant SLAs when allocating the resources. For the one-tier systems, only that proposed in [19] achieves both dynamic resource allocation and latency compliance. However, as described above, the system focuses mainly on the edge resources. Moreover, among the two-tier systems, none of the proposed methods support both dynamic resource allocation and latency compliance.

Therefore, due to the inadequate solutions offered by previous works, we propose an efficient method that addresses critical issues in 5G. Our method incorporates three key properties: (1) real-time resource utilization of network nodes for dynamic resource allocation and task offloading optimization; (2) reduced service request blocking rates through dynamic resource allocation for optimal resource availability; and (3) optimized resource allocation for each tenant network slice to enhance overall efficiency and effectiveness of the multi-tenant 5G network, and improve service quality for tenants. Specifically, our approach is based on the collaboration of two algorithms (MCRA and FLDRA) and three models (*CO*, *EO*, *PH*). These algorithms enable dynamic path routing and resource allocation that satisfies the latency constraints of all tenant SLAs while maximizing resource utilization and minimizing the probability of access delays and slice blocking.

## 3. System Architecture

### 3.1. System Model

The aim of the proposed network structure and resource allocation algorithm in this study is to target the shortcomings of related works and to solve the three deficiencies identified by Chien et al. [20]. The steps taken in this study to target these deficiencies can be briefly described as follows:Chien et al. [20] did not consider the core network. Thus, the present study defines a two-tier architecture that takes explicit account of this network.In [20], the Transport Network (TN) was considered a component of the overall architecture and had its own queue. However, in reality, the TN can be modeled as a simple communication queue compared to the core network, since it is a transmission environment that consists of multiple network devices. Therefore, the TN is redundant, and in our proposed architecture, we have removed it.In [20], the RAN and edge server are separated into different locations within the architecture. However, in real 5G network environments, the edge servers coexist with the RAN at the base station. As a result, in the architecture proposed in this study, the communication resources and queueing model between the RAN and edge server are ignored.Additionally, in [20], the resource allocation tasks for different service requests are handled separately. In other words, the allocated resource for each service request is based simply on that assigned to the first request. That is, no provision is made for adjusting the allocated resources dynamically across different service requests based on changes in the available resources at the nodes. In our proposed algorithm, we aim to dynamically allocate the given resources to each request. This involves adjusting the allocation of resources in response to changes in available resources at the nodes. By performing dynamic allocation, our algorithm can more effectively and efficiently Satisfy the resource requirements of each request, ultimately improving the overall performance of the system.In implementing the service diversion strategy in [20], the diversion ratio on different nodes is simply increased or decreased in increments of 1% until the resource limit is reached. However, this approach does not consider the relationship between the cost and the E2E delay, or the resource allocation ratios under each slice. Thus, the present study proposes a resource allocation strategy which optimizes the tradeoff between the cost weight and the minimum delay in each slice when adjusting the diversion ratio at the nodes.

Figure 1 shows the two-tier architecture proposed in the present study to address the limitations of [20] described above. As shown, the architecture consists mainly of a CN, several central offices, and multiple Access Networks (ANs). The CN provides computing functions, including MEC, and is paired with the central offices located at distances of up to 100 km away. In traditional networks, the central office end-to-UE link only has access to the network. By contrast, in the architecture shown in Figure 1, the ANs are treated as the edge and add MEC to provide an edge computing capability.

Figure 2 presents a simplified view of the proposed two-tier architecture for real-world scenarios in which multiple User Equipments (UEs) are attached to eachNext Generation Node B (gNB) and the central office serves only as a network component for packet forwarding. Any UE with a task to perform sends a request to the gNB through the RAN, and the gNB then decides where the corresponding MEC computation is to be performed (i.e., at the edge or through the CN to the cloud server).

Given the architecture show in Figure 2, the packet flow between the various network components can be visualized as shown in Figure 3. As shown, the network’s computing resources include C and E, while its communication resources consist of RAN and core network.

For 5G systems, the two-tier architecture shown in Figure 3 can support multiple tenants, each of which has three slices, where each slice corresponds to a particular service type, such as eMBB, URLLC, and mMTC, with its own QoS or SLA requirements. In other words, each slice has its own dedicated computation and communication resources. It can be assumed that the service requests for each slice follow a Poisson process, and each request has its own computational and communication requirements together with a required service time. It can be further assumed that the service times of the requests follow an exponential distribution.

In practice, service requests within the same network slice have the same service type and SLA/QoS requirements. Therefore, all of the service requests share the resources allocated to that slice. In other words, in the system considered in the present study, the individual service requests do not receive dedicated resources allocated specifically to them (as in [20]), but rather share the resources of the slice in such a way as to ensure a consistent service quality.

To optimize the provision of computing and communication resources to the different tenants of different services under 5G network slicing, the present study considers three service request models based on the two-tier architecture, namely *CO*, *EO*, and *PH*. In the *CO* and *EO* models, the service requests are served solely by the C or E, respectively. By contrast, in the *PH* model, the service requests are assigned dynamically to either the edge or the cloud server based on the optimal allocation of the computing and communication resources.

### 3.2. Problem Definition

In the present study, the goal is to optimize the resource allocation and service capacity of each network slice while simultaneously satisfying the service quality requirements of the tenants. To facilitate this goal, the queueing network within the two-tier architecture is designed with the form shown in Figure 4 and comprises four communication queues, $T_{i,l}^{\mathrm{RU}}$, $T_{i,l}^{\mathrm{RD}}$, $T_{i,l}^{\mathrm{CU}}$, and $T_{i,l}^{\mathrm{CD}}$, corresponding to the uplink and downlink queues from the UE to the edge server and the edge server to the cloud server, respectively. The network additionally includes two computation queues, $T_{i,l}^{E}$ and $T_{i,l}^{C}$, corresponding to the edge server and cloud server, respectively. To facilitate the following discussions, Table 2 lists the notations used throughout the remainder of this paper.

Initially, the upper usage limits of each resource in the network slice are set as $MR_{i,l}^{C}$, $MR_{i,l}^{E}$, $MR_{i,l}^{\mathrm{CU}}$, $MR_{i,l}^{\mathrm{CD}}$, $MR_{i,l}^{\mathrm{RU}}$, and $MR_{i,l}^{\mathrm{RD}}$, where C and E represent the cloud and edge, respectively, and CN Uplink (CU), CN Downlink (CD), RN Uplink (RU), and RN Downlink (RD) represent the uplinks and downlinks of the CN and RN, respectively.

Regarding the tenant leasing information, the *j* th service request from the *i* th tenant in the *l* th slice is represented as $r_{i,j,l}$. The service request information includes the tolerable average delay of the service $D_{i,l}$ and the estimated traffic arrival rate $\lambda_{i,j,l}$. In addition, each tenant has a guaranteed quality of service $SLA_{i,l}$, which specifies the amount of guaranteed resources required for each queue. In the problem considered in this study, the service requests of the tenants are translated into resource requirements, and the slice resources are allocated to each tenant as $R_{i,j,l}^{\mathrm{RU}}$, $R_{i,j,l}^{\mathrm{RD}}$, $R_{i,j,l}^{\mathrm{CU}}$, $R_{i,j,l}^{\mathrm{CD}}$, $R_{i,j,l}^{C}$, and $R_{i,j,l}^{E}$., respectively. The problem can be formally defined as follows:Inputs: The two-tier architecture (C, E, CN, RN, UE); the number of network slices supported (*L*); the tenants and their SLAs (*T*, *S*, SLA, *D*, VNF); the service requests from the tenants (Λ, M, X, *r*, λ,μ); and the *j* th service request from the *i* th tenant in the *l* th slice ($r_{i,j,l}$).Output: The resources required by each node to satisfy the current request ($R_{i,j,l}^{\mathrm{RU}}$, $R_{i,j,l}^{\mathrm{RD}}$, $R_{i,j,l}^{\mathrm{CU}}$, $R_{i,j,l}^{\mathrm{CD}}$, $R_{i,j,l}^{C}$, $R_{i,j,l}^{E}$).Constraints:The specific resources of each tier (C, E, CN, RN, UE) required by each tenant must not exceed the corresponding SLA ($S_{i,l}$) resource allocation limit.The calculated E2E delay must satisfy the corresponding delay constraint for each tenant slice ($D_{i,l}$).**Objective:** Maximize the system resource availability $\begin{array}{c} \sum_{i,l}SA_{i,l} \end{array}$.

## 4. Proposed Methodology

### 4.1. Resource Allocation Work Flow

Figure 5 presents a flowchart of the dynamic resource allocation method proposed in the present study. When a new service request arrives, it is assigned a small initial resource demand, and the resulting latency, which is derived from this resource demand, is compared with the latency constraint of the corresponding service. If the constraint is satisfied, the corresponding resource requirements are computed; otherwise, two newly-proposed algorithms, namely MCRA or FLDRA, are applied as appropriate to increase the required resources until the latency constraint is satisfied. The required resources are then compared with the remaining resources in the slice. If the residual resources are sufficient, the system allocates the resources in line with the resource requirements; otherwise, the system blocks the request.

### 4.2. Latency Modeling

In the resource allocation procedure described above, each service node is modeled as an M/M/1 queue.M/M/1 is a notation used in Kendall’s queueing systems, in which “M” denotes a Poisson arrival process. Specifically, this implies that the service facility receives requests that arrive independently and with equal probability, and the inter-arrival times between requests are exponentially distributed. The second “M” in “M/M/1” refers to the distribution of service times, which is also exponential. This indicates that the service time follows an exponential distribution, with a mean of $\frac{1}{\mu }$. Lastly, the “1” in the notation indicates that the system comprises only one service node, which implies that each request can be served by only one service node at any given time. In this paper, the service nodes include both computation nodes, such as the C and E, and communication links, such as the RU, RD, CU, and CD. Each communication link is assumed to have both a propagation delay and an M/M/1 system delay, where the propagation delay is a fixed constant. The M/M/1 system delay includes the transmission delay of the queue and the waiting time in the queue, which follows an exponential distribution such as μ - λ, where μ is the communication resource demand and λ is the traffic arrival rate. The average delay for the M/M/1 queue at the computation nodes (E and C) is thus equal to $\frac{1}{\mu -\lambda }$.

The present study considers three different network models: *CO*, *EO*, and *PH*. In the *CO* model, all of the traffic is handled by the central cloud via one server and four communication links: C, CU, CD, RU, and RD. Note that in this case, the gNB is considered as the edge and does not provide any services; hence, it is not considered in the model. In the *EO* model, all of the traffic is handled by the edge (gNB), and the services are provided by just one server and two communication links: E, RU, and RD. In the *PH* model, each request is dynamically assigned to specific servers (E and C) and communication links (CU, CD, RU, and RD) based on the current resource utilization, allowing for efficient processing of each service.

In the *CO* and *EO* models, the propagation delay is influenced by the propagation speed of the signal through the wire or optical fiber, which is around two-thirds the speed of light, as well as the distance. When an incoming request ($r_{i,j,l}$) is received, its one-way propagation delay ($D_{i,l}^{P_{m}}$) is defined by Equations ([Disp-formula FD1a-sensors-23-04698]), ([Disp-formula FD1b-sensors-23-04698]) and ([Disp-formula FD1c-sensors-23-04698]) as follows:
(1a)$$\begin{array}{c} D_{i,l}^{P_{m}}=\frac{3\times distance}{2\times lightspeed} \end{array}$$
(1b)$$\begin{array}{c} m\in \{CO,EO\} \end{array}$$
(1c)$$\begin{array}{c} distance=\left\{\begin{array}{cc} Dist^{UE2gNB}+Dist^{gNB2C}, & \mathrm{if}\;m=CO.\\ Dist^{UE2gNB}, & \mathrm{otherwise}. \end{array}\right. \end{array}$$

The round trip propagation delay is thus defined as:(2)$$2\times D_{i,l}^{P_{m}}=\frac{3\times distance}{lightspeed}$$

In the *CO* model, the uplink transmission delay ($D_{i,j,l}^{\mathrm{NU}}$) is given by the sum of the transmission times and queueing delays at RU ($D_{i,j,l}^{\mathrm{RU}}$) and CU ($D_{i,j,l}^{\mathrm{CU}}$). By contrast, in the *EO* model, the uplink transmission delay is equal simply to the transmission delay at RU ($D_{i,j,l}^{\mathrm{RU}}$). Mathematically, the two transmission delays can thus be expressed as:
(3a)$$\begin{array}{c} D_{i,j,l}^{u}=\frac{1}{\frac{\left(F_{i,l}^{u}+R_{i,j,l}^{u}\right)}{\mu_{i,j,l}^{\mathrm{NU}}-\left({\tilde{\lambda }}_{i,l}+\lambda_{i,j,l}\right)}} \end{array}$$
(3b)$$\begin{array}{c} D_{i,j,l}^{\mathrm{NU}}=\left\{\begin{array}{cc} \sum_{u\in \{\mathrm{RU},\mathrm{CU}\}}D_{i,j,l}^{u}, & \mathrm{if}\;m=CO.\\ \sum_{u\in \{\mathrm{RU}\}}D_{i,j,l}^{u}, & \mathrm{otherwise}. \end{array}\right. \end{array}$$

The downlink transmission delay ($D_{i,j,l}^{\mathrm{ND}}$) in the *CO* model is again equal to the sum of the transmission time and queueing delays at RD ($D_{i,j,l}^{\mathrm{RD}}$) and CD ($D_{i,j,l}^{\mathrm{CD}}$). Similarly, in the *EO* model, the downlink transmission delay is equal simply to RD ($D_{i,j,l}^{\mathrm{RD}}$). The two downlink transmission delays can thus be expressed mathematically as:
(4a)$$\begin{array}{c} D_{i,j,l}^{d}=\frac{1}{\frac{\left(F_{i,l}^{d}+R_{i,j,l}^{d}\right)}{\mu_{i,j,l}^{\mathrm{ND}}-\left({\tilde{\lambda }}_{i,l}+\lambda_{i,j,l}\right)}} \end{array}$$
(4b)$$\begin{array}{c} D_{i,j,l}^{\mathrm{ND}}=\left\{\begin{array}{cc} \sum_{d\in \{\mathrm{RD},\mathrm{CD}\}}D_{i,j,l}^{d}, & \mathrm{if}\;m=CO.\\ \sum_{d\in \{\mathrm{RD}\}}D_{i,j,l}^{d}, & \mathrm{otherwise}. \end{array}\right. \end{array}$$

Overall, the E2E network delay ($D_{i,j,l}^{Net_{m}}$) and computational delay ($D_{i,j,l}^{Comp_{m}}$) can be computed as shown in Equations ([Disp-formula FD5a-sensors-23-04698]) and ([Disp-formula FD5b-sensors-23-04698]), respectively. The overall E2E delay ($D_{i,j,l}^{E2E_{m}}$) is then given by the sum of the E2E network delay and the computational delay, as shown in Equation ([Disp-formula FD5c-sensors-23-04698]). Note that m is defined in Equation ([Disp-formula FD1b-sensors-23-04698]), which indicates whether the model is a *CO* model or an *EO* model.  
(5a)$$\begin{array}{c} D_{i,j,l}^{Net_{m}}=2\times D_{i,l}^{P_{m}}+D_{i,j,l}^{\mathrm{NU}}+D_{i,j,l}^{\mathrm{ND}} \end{array}$$
(5b)$$\begin{array}{c} D_{i,j,l}^{Comp_{m}}=D_{i,j,l}^{m}=\frac{1}{\frac{\left(F_{i,l}^{m}+R_{i,j,l}^{m}\right)}{\mu_{i,j,l}^{m}-\left({\tilde{\lambda }}_{i,l}+\lambda_{i,j,l}\right)}} \end{array}$$
(5c)$$\begin{array}{c} D_{i,j,l}^{E2E_{m}}=D_{i,j,l}^{Comp_{m}}+D_{i,j,l}^{Net_{m}} \end{array}$$

## 5. Algorithm Designs

This section commences by describing the processes used to determine the resource allocation for each service request in the *CO*, *EO* and *PH* models, respectively. The MCRA and FLDRA algorithms used to allocate resource units to the nodes with a lower cost or a minimum latency, respectively, are then introduced.

### 5.1. Determine Resource Allocation for Each Request in Cloud-Only and Edge-Only Models

As shown in Algorithm 1, the process of determining the resource allocation (Ri,j,lX) required to serve a new request commences with the decision as to whether this request should be processed using the *CO* model or the *EO* model. Having chosen the appropriate model, an initial resource allocation is determined and an appropriate node X chosen. If the minimum resource allocation and latency constraints are both satisfied, the request is serviced at the chosen node using the determined resource allocation; else, the MCRA or FLDRA algorithm is used to allocate resource to make the overall E2E delay (Di,j,lE2Em) satisfy the delay constraint.
**Algorithm 1** Resource allocation determination for each request in Cloud-only (*CO*) and Edge-only (*EO*) Model**Input:** m,Fi,lX,Ri,j,lX,λ˜i,l,λi,j,l,μi,j,lX,Di,l**Output:** Ri,j,lX  1:**if** m=CO **then**  2:    Ri,j,lX←λi,j,l×μi,j,lX,X∈{C,RU,RD,CU,CD}  3:**else**  4:    Ri,j,lX←λi,j,l×μi,j,lX,X∈{E,RU,RD}  5:**end if**  6:Compute Di,j,lE2Em using Equation ([Disp-formula FD5c-sensors-23-04698])  7:**if** Fi,lX+Ri,j,lX>(λ˜i,l+λi,j,l)×μi,j,lX and Di,j,lE2E_m≤Di,l **then**  8:    Ri,j,lX←λi,j,l×μi,j,lX  9:**else if** Fi,lX+Ri,j,lX>(λ˜i,l+λi,j,l)×μi,j,lX and Di,j,lE2Em>Di,l **then**10:    Ri,j,lX←XRA(m,Di,j,lE2Em,Di,l,μi,j,lX,∂,Fi,lX,Ri,j,lX,MRi,lX) {XRA is MCRA or FLDRA}11:**end if**

### 5.2. Determine Resource Allocation for Each Request in Parallel Cloud-Edge Hybrid Model

The basic concept of the *PH* model is to select the model (*CO* or *EO*) which best satisfies the service request with a resource allocation (Ri,j,lX) adaptively based on the present resource allocation and latency conditions in the slice. In particular, if the minimum allocation constraint is satisfied but the latency constraint is not, the MCRA algorithm is used to calculate the costs of the C and E (Costi,j,lPHC and Costi,j,lPHE), respectively, and to then select the model which returns the lowest cost to serve the request. Alternatively, the FLDRA algorithm is used to calculate the delays of the C and E (Di,j,lPHC and Di,j,lPHE), respectively, and the model with the lowest delay is then chosen to serve the request. The corresponding pseudo-code is presented in Algorithm 2.
**Algorithm 2** Resource allocation determination for each request in Parallel Cloud-Edge Hybrid (*PH*) Model**Input:** m,Fi,lX,Ri,j,lX,λ˜i,l,λi,j,l,μi,j,lX,Di,j,lE2Em,Di,l,k,γX**Output:** Ri,j,lX  1:**if** Fi,lX+Ri,j,lX>(λ˜i,l+λi,j,l)×μi,j,lX & Di,j,lE2Em≤Di,l **then**  2:    Ri,j,lX←λi,j,l×μi,j,lX  3:**else if** Fi,lX+Ri,j,lX>(λ˜i,l+λi,j,l)×μi,j,lX & Di,j,lE2Em>Di,l **then**  4:    Employ XRA in *CO* {XRA is MCRA or FLDRA}  5:    Costi,j,lPHC←∑X∈{C,RU,RD,CU,CD}γX  6:    **for** each node X **do**  7:        Ri,j,lPHC_X←Ri,j,lX+(α×μi,j,lX);  8:    **end for**  9:    Employ XRA in *EO* {XRA is MCRA or FLDRA}10:    Costi,j,lPHE←∑X∈{E,RU,RD}γX11:    **for** each node X **do**12:        Ri,j,lPHE_X←Ri,j,lX+(α×μi,j,lX);13:    **end for**14:    **if** Costi,j,lPHC<Costi,j,lPHE **then**15:        **for** each node X **do**16:            Ri,j,lX←Ri,j,lPHC_X;17:        **end for**18:    **else**19:        **for** each node X **do**20:            Ri,j,lX←Ri,j,lPHE_X;21:        **end for**22:    **end if**23:**end if**

### 5.3. Minimum Cost Resource Allocation Algorithm

Algorithm 3 shows the pseudo-code of the MCRA algorithm. In situations where the E2E latency constraint is not satisfied, MCRA prioritizes the allocation of resource units to the node with the lowest cost under the maximum capacity constraint. In general, the nodes in the network comprise either computational resources or communication resources. When allocating the initial resources, Ri,j,lX=λi,j,l×μi,j,lX, the smallest resource unit is set as μi,j,lX, and the smallest resource unit at the different nodes is then determined as α×μi,j,lX. For computational resources, α×μi,j,lX is set to 1, while for communication resources, it is set to 0.1×μi,j,lX. For example, assuming that node X has the lowest cost, if Fi,lX+Ri,j,lX+(α×μi,j,lX)≤MRi,lX, then Ri,j,lX←Ri,j,lX+(α×μi,j,lX). Based on the on-line routing concept in [30], the cost of allocating one unit of resource at node X is given by:(6)γX=∂(Fi,lX+Ri,j,lX+(α×μi,j,lX)MRi,lX−1)−∂(Fi,lX+Ri,j,lXMRi,lX−1).
**Algorithm 3** Minimum Cost Resource Allocation (MCRA) algorithm**Input:** m,Di,j,lE2Em,Di,l,μi,j,lX,∂,Fi,lX,Ri,j,lX,MRi,lX**Output:** Ri,j,lX  1:**if** m=CO **then**  2:    X∈{C,RU,RD,CU,CD}  3:**else**  4:    X∈{E,RU,RD}  5:**end if**  6:**while** Di,j,lE2Em>Di,l **do**  7:    **if** X∈{C,E} **then**  8:        α←1/μi,j,lX  9:    **else**10:        α←0.111:    **end if**12:    X←argminXγX, where γX=∂(Fi,lX+Ri,j,lX+(α×μi,j,lX)MRi,lX−1)−∂(Fi,lX+Ri,j,lXMRi,lX−1)13:    **if** Fi,lX+Ri,j,lX+(α×μi,j,lX)≤MRi,lX **then**14:        Ri,j,lX←Ri,j,lX+(α×μi,j,lX);15:    **else**16:        Ri,j,lX←017:        exit with error (not enough resource)18:    **end if**19:    Compute Di,j,lE2Em using Equation ([Disp-formula FD5c-sensors-23-04698])20:**end while**

We will now analyze the time complexity of the MCRA algorithm. First, we determine the complexity for acquiring the node at minimum cost, which is equivalent to the size of X, denoted by $\left|X\right|$. From lines 2 and 4, we can observe that $\left|X\right|$ is less than or equal to 5. Thus, it can be considered negligible. Second, in the while loop between lines 6 and 20, the pseudo-code requires allocating $\Delta_{i,j,l}$ units of resource to each node with a unit size of $\alpha \times \mu_{i,j,l}$, where $\Delta_{i,j,l}$ depends on the delay constraint and $\alpha \times \mu_{i,j,l}$ is greater than 1. In the worst case scenario, the resource allocation is bounded by $MR_{i,l}^{X}$, which means it may take $MR_{i,l}^{X}$ iterations. Therefore, the time complexity of the MCRA algorithm is $O(\left|X\right|\times MR_{i,l}^{X})$, but since $\left|X\right|$ is negligible, we can reduce it to $O(MR_{i,l}^{X})$.

### 5.4. Fast Latency Decrease Resource Allocation Algorithm

Algorithm 4 shows the pseudo-code of the FLDRA algorithm. In contrast to the MCRA algorithm, which selects the node with the lowest cost for resource allocation, FLDRA selects the node which yields the greatest reduction in the E2E delay for resource allocation. Note that the reduction in delay caused by a given resource on node X is evaluated as: (7)DunitX=1Fi,lX+Ri,j,lX+λi,j,l×μi,j,lXμi,j,lX−(λ¯i,lX+λi,j,l)−1Fi,lX+Ri,j,lX+α×μi,j,lXμi,j,lX−(λ¯i,lX+λi,j,l)
**Algorithm 4** Fast Latency Decrease Resource Allocation (FLDRA) algorithm**Input:** m,Di,j,lE2Em,Di,l,μi,j,lX,∂,Fi,lX,Ri,j,lX,MRi,lX**Output:** Ri,j,lX  1:**if** m=CO **then**  2:    X∈{C,RU,RD,CU,CD}  3:**else**  4:    X∈{E,RU,RD}  5:**end if**  6:**while** Di,j,lE2Em>Di,l **do**  7:    **if** X∈{C,E} **then**  8:        α←1/μi,j,lX  9:    **else**10:        α←0.1.11:    **end if**12:    X←argmaxXDunitX13:    **if** Fi,lX+Ri,j,lX+(α×μi,j,lX)≤MRi,lX **then**14:        Ri,j,lX←Ri,j,lX+(α×μi,j,lX);15:    **else**16:        Ri,j,lX←017:        exit with error (not enough resource)18:    **end if**19:    Compute Di,j,lE2Em using Equation ([Disp-formula FD5c-sensors-23-04698])20:**end while**

The time complexity of the FLDRA algorithm is the same as that of MCRA, since the only difference between them is the selection criterion for nodes (line 12). Therefore, the time complexity of the FLDRA algorithm is $O(MR_{i,l}^{X})$.

## 6. Simulation

This section commences by describing the simulation environment used to evaluate the performance of the proposed dynamic resource allocation schemes. The simulation results are then presented and discussed.

### 6.1. Simulation Environment

The simulation environment was built using C programming language on a PC with 24 GB of RAM and the Windows operating system. The simulations considered two tenants ($S_{i,l},i=1,2$) with three slices (l=1,2,3) (eMBB, URLLC, and mMTC). The configurations of parameters varied depending on the particular slice applications, which were defined as follows:Tenant 1:eMBB: High data rates and traffic density performance criteria as Urban macroURLLC: Process automation–remote controlmMTC: Low data rate and maintaining battery lifeTenant 2:eMBB: 4K 360-degree VR panoramic video (VR IMAX and VR concert)URLLC: Intelligent transport systems–infrastructure backhaulmMTC: Medical monitoring

Table 3 shows the parameter settings for each slice of Tenant 1 (T1) and Tenant 2 (T2). In general, the settings configure the latency constraints, resource demands, and guaranteed resources for the different slices of each tenant. For the eMBB slice, the latency constraints of T1 and T2 are specified as 500 ms [31] and 100 ms [32], respectively. For the URLLC slice, the corresponding latencies are 60 ms [33] and 30 ms [33], respectively. Finally, for the mMTC slice, the latencies are set as 10,000 ms [31] and 50 ms [34].

For T1, the uplink and downlink resource demands for the eMBB slice are set as 2.5 Mbps and 5 Mbps, respectively, while those for the URLLC slice are 20 Mbps and 1 Mbps [35], respectively, and those for the mMTC slice are 0.14 Mbps and 0.125 Mbps [36]. For T2, the uplink and downlink resource demands for the eMBB, URLLC and mMTC slices are 4 Mbps and 8 Mbps [32], 2 Mbps and 0.2 Mbps [35], and 0.1 Mbps and 0.05 Mbps [34], respectively.

For both tenants, and each slice, the unit resource usage is equal to the product of the resource demand and the packet arrival rate. For the eMBB slice, the traffic arrival rate is set equal to 10 for both T1 and T2. Thus, the maximum uplink and downlink rates for T1 are 25 Mbps and 50 Mbps, respectively [33], while those for T2 are 40 Mbps and 80 Mbps. Note that a similar calculation process can be applied to the other four slices.

### 6.2. Simulation Results

The simulation experiments considered three network structure models: *CO*, *EO*, and *PH*. The MCRA and FLDRA algorithms were designed using various resource allocation methods and diversion strategies. The simulations compared the E2E delay, the probability of service request blocking, and the remaining resource availability at each node. The simulations additionally investigated the impact of changes in the service request rate. Finally, the performance of the proposed algorithms and network architectures were compared with that of the UFLOP algorithm and architecture proposed in [20].

This section commences by comparing the performance of the MCRA and FLDRA algorithms in the *CO*, *EO*, and *PH* models, respectively (Section 6.2.1, Section 6.2.2 and Section 6.2.3). The performance of MCRA algorithm is then compared with that of UFLOP [20] in the three models, respectively (Section 6.2.4, Section 6.2.5 and Section 6.2.6). Finally, the request blocking rates of MCRA, FLDRA and UFLOP are evaluated and compared for the three different network models (Section 6.2.7).

#### 6.2.1. Performance of MCRA and FLDRA Algorithms in Cloud-Only Model

Average DelayFigure 6 shows the average delays of the MCRA and FLDRA algorithms in the different slices of the two tenants in the *CO* model. Note that the horizontal axis shows the request arrival rate of the entire system, while the vertical axis shows the average E2E delay. The results show that, through an appropriate incremental adjustment of the resources ($R_{i,j,l}^{X}=R_{i,j,l}^{X}+alpha*\mu_{i,j,l}^{X}$), both algorithms enable the E2E delay constraints of the three slices to be satisfied for both tenants.Although both algorithms satisfy the SLA of each slice, the average delay of the MCRA algorithm is less than that of the FLDRA algorithm. This finding is reasonable since the FLDRA algorithm selects the node which results in the largest delay drop (as computed by Table 3, Equations ([Disp-formula FD3a-sensors-23-04698]) and ([Disp-formula FD4a-sensors-23-04698]), and line 10 of the Algorithm 4). Consequently, the delay gradually decreases with each adjustment. When the average delay satisfies the SLA, the system will stop adjusting, resulting in the node being very close to the SLA. By contrast, the MCRA algorithm calculates weights based on each node’s current resource usage and allocates resources to nodes with more remaining resources (as shown in Equation (Equation 6)). Therefore, when the algorithm selects the node with the lowest cost, it will inevitably cause the average delay to be lower than the SLA, and even significantly lower than the SLA, resulting in a greater reduction of the E2E delay.Resource AvailabilityFigure 7a,d shows the available resources at each node in the eMBB slices of the two tenants when applying the MCRA and FLDRA algorithms, respectively, and using the *CO* model. The horizontal axis represents the request arrival rate of the entire system, while the vertical axis represents the remaining resources at each node. A detailed analysis shows that the standard deviation of the available resources at each node under different arrival rates is around 0.01–0.05 for the MCRA algorithm and 0.2–0.35 for the FLDRA algorithm. In other words, the MCRA algorithm results in a more uniform utilization of the node resources in the eMBB slice than the FLDRA algorithm for each tenant.In the eMBB slice, the required resources and latency constraints of the T1 requests are smaller than those of T2. However, the T1 requests have a longer survival time (T1 = 5000 ms vs. T2 = 1000 ms, Table 3), and consequently, the resource utilization of T1 is greater than that of T2. From the perspective of the number of resource allocation adjustments, FLDRA selects nodes which yield the greatest reduction in the E2E delay. Hence, each node has been adjusted approximately the same number of times. However, due to the tenfold difference in bandwidth between the core network and RN (i.e., $MR_{i,l}^{\mathrm{CU}}$ is 10 times that of $MR_{i,l}^{\mathrm{RU}}$), the remaining available capacity at the RN node is relatively small, while that of the core network is relatively large. In addition, for T1 (high data rate and traffic density), the remaining available capacity of the C computing resource is greater than that of any of the other nodes for all values of the traffic arrival rate, while for T2 (360-degree VR panoramic video), the remaining available capacity of the C computing resource is the lowest of all the nodes.Figure 7b,e shows the available resources at each node in the URLLC slices of the two tenants under the two algorithms and the *CO* model. For both tenants, the standard deviation of the available resources under the MCRA algorithm is around 0.01–0.05, while for the FLDRA algorithm, the standard deviation is around 0.2–0.3. In other words, for both tenants, the MCRA algorithm achieves a more balanced resource utilization across the URLLC slice since it favors nodes with the lowest cost when allocating resources, where this cost is calculated based on the usage of the nodes. In particular, as shown in Algorithm 3. $\gamma^{X}$ is calculated based on *∂*, and the exponential term of *∂* is divided by $MR_{i,l}^{X}$, which can be interpreted as a proportional adjustment. By contrast, in the FLDRA algorithm, the resource allocation is adjusted based on the delay, and hence the resource usage across the different nodes is relatively unbalanced.In the URLLC slice, T1 has a higher demand for single-request resources than T2, resulting in a relatively higher occupancy of the resources. Although T2 has relatively lower latency constraints, the impact of this on the resource availability is less significant than that of the resource demands of T1. As a result, the availability of the computational resources of T2 is greater than that of T1. In general, due to the strict latency requirements of vehicle-to-vehicle communications, the computational resources allocated to the URLLC slice are slightly higher than those allocated to the other slices, resulting in fewer remaining RN resources and more remaining CN resources. Notably, for T1 (remote control), the remaining available resources at the C computing node are the lowest among all the nodes, while for T2 (infrastructure backhaul), the remaining C computing resources fall between those of the RN and CN, respectively.Figure 7c,f shows the available resources at each node of the mMTC slices when using the two algorithms and the *CO* model. The standard deviation of the available resources of the different nodes when using the MCRA algorithm is approximately 0.01–0.03. By contrast, that for the FLDRA algorithm is round 0.3–0.4. In other words, as for the eMBB and URLLC slices, the MCRA algorithm results in a more uniform resource utilization than the FLDRA algorithm for both tenants.In the mMTC slice, the latency constraint of T1 is much larger than that of T2 (T1 = 10,000 ms vs. T2 = 50 ms). Thus, it seems intuitive that T1 should require fewer resources to service each request. In Figure 7c, the application of T1 is for maintaining low data rates and battery life. Therefore, in the initial resource design, more resources are provided to the computing nodes and Core Network compared to RN, and requests are assigned to the computing nodes and CN for processing to reduce the frequency of discharging and charging, thereby extending the battery life. As a result, blocking only occurs when the arrival rate reaches 150. In Figure 7f, T2 is used for medical monitoring, and the communication nodes have more initial resources than the computing nodes. However, due to the FLDRA adjustment mechanism, each node experiences a relatively uniform number of adjustments. As a result, the T2 computing nodes with fewer initial resources will experience a reduction in resource availability after several allocations, resulting in a blocking phenomenon occurring at 60 (as shown in Figure 8c).Blocking RateFigure 8 shows the blocking rates of the two tenants for the different slices when using the MCRA and FLDRA algorithms in the *CO* model. The horizontal axis represents the request arrival rate of the entire system, while the vertical axis represents the blocking rate of each slice. In the simulations, the service requests are evenly distributed among the six slices, and hence the blocking rate rises relatively slowly in each case. For all six slices, the blocking rate caused by FLDRA is higher than that caused by MCRA. In the mMTC slice, the low latency constraint of 50 ms of T2 results in blocking even at low arrival rates. Due to its policy of selecting the node which yields the most significant reduction in the E2E delay, the blocking rate of T2 under FLDRA increases significantly as the request arrival rate increases beyond 60. Overall, the results show that, for both algorithms, a stricter latency requirement has a greater impact on the blocking rate, particularly at higher request arrival rates.

#### 6.2.2. Performance of MCRA and FLDRA Algorithms in Edge-Only Model

Average DelayFigure 9 shows the average delays in the three slices of the two tenants when using the MCRA and FLDRA algorithms under the *EO* model. The results are similar to those obtained for the *CO* model. In particular, when the initial E2E delay exceeds 1 s, the resource allocation is adjusted by the MCRA or FLDRA algorithm in such a way as to satisfy the respective latency constraints of the various applications. Both algorithms satisfy the SLA requirements of the two tenants in every slice. However, as for the *EO* model, the average delay under the MCRA algorithm is lower than that under FLDRA, particularly in the mMTC slice of T1. This is because the T1 mMTC slice has a much greater latency constraint (10,000 ms) compared to the other slices, which results in the significant difference in average latency between the two algorithms.Resource AvailabilityFigure 10a,d shows the availability of the resources at each node in the eMBB slices of the two tenants when using the MCRA and FLDRA algorithms and the *EO* model. For the MCRA algorithm, the standard deviation of the resource availability across the different nodes ranges from 0.01 to 0.05. By contrast, that for the FLDRA algorithm ranges from 0.15 to 0.3. In other words, MCRA results in a more uniform usage of the node resources than FLDRA for both tenants. T1 has lower resource and latency requirements than T2 in the eMBB slice. However, its service requests have a longer survival time (i.e., 5000 ms compared to 1000 ms for T2). Consequently, the resource usage is similar for both tenants in the eMBB slice. However, a detailed inspection reveals that T1 has slightly fewer remaining transmission resources than computational resources, while T2 has slightly fewer remaining computational resources. Finally, the difference in the amount of remaining available resources among the different nodes in the FLDRA algorithm is greater than that under the MCRA algorithm. This can be attributed to the fact that the CN resources are 10 times greater than that of the RN. In FLDRA, the mechanism results in the number of adjustments for each node being almost equal, without giving priority to nodes with more remaining resources for further adjustments. As a result, significant differences in remaining resource availability among nodes occur. By contrast, in MCRA, the system calculates weights based on each node’s current resource usage and allocates resources to nodes with more remaining resources (as shown in Equation (Equation 6)). Therefore, the mechanism of MCRA, resulting in different adjustment numbers for each node, promotes a more even distribution of available remaining resources.Figure 10b,e shows the resource availability at each node in the URLLC slices of the two tenants under the MCRA and FLDRA algorithms and *EO* model. The standard deviation of the resource availability for MCRA is approximately 0.01–0.05, while that for FLDRA it is around 0.15–0.2. In other words, for both tenants, MCRA achieves a more balanced distribution of the resource utilization of the nodes in the slice than FLDR. Although T1 demands more resources than T2 to satisfy its service requests in the URLLC slice, the latency constraints of T2 are more relaxed than those of T1, and hence the resource utilization of T2 is greater than that of T1. Under the MCRA and FLDRA algorithm, the remaining RN transmission resources and computational resources for T1 are more than those for T2. In each slice, the difference in the remaining available resources at the different nodes in the *EO* model is greater under the FLDRA algorithm than under the MCRA algorithm.Figure 10c,f shows the resource availability at each node of the mMTC slice of the two tenants when using the two resource allocation algorithms and the *EO* model. The standard deviation of the resource availability under MCRA is approximately 0.03–0.08, while that under FLDRA it is around 0.1–0.25. In other words, MCRA once again shows a relatively more balanced utilization of the node resources than FLDRA for both tenants.In the mMTC slice, T1 requires fewer resources to service its requests, but has a longer survival time. Consequently, the remaining resource availability reduces rapidly as the request arrival rate increases. Conversely, T2 requires more resources to service its requests. As a result, blocking occurs when the resource availability reduces to around 15% (see Figure 10f). In the MCRA algorithm, T1 has more remaining RN transmission resources and computational resources than T2. For both tenants, the disparity in the available resources among the different nodes is greater under the FLDRA algorithm than under the MCRA algorithm.Blocking RateFigure 11 shows the blocking rates of the MCRA and FLDRA algorithms for the different slices of the two tenants under the *EO* model. FLDRA results in a higher blocking rate than MCRA across all six slices, and for the URLLC and mMTC slices in particular. This outcome can be attributed to two main factors. First, the *EO* model has only three nodes, compared to the five nodes in the *CO* model, which exacerbates the uneven resource usage tendency of FLDRA and increases the number of necessary resource adjustments, especially for nodes with abnormally high resource usage. Second, computational resources are scarcer on the edge than in the C, which increases the likelihood of resource shortages for requests that require a large amount of computational resources. As a result, the service requests are more likely to be blocked.

#### 6.2.3. Performance of MCRA and FLDRA Algorithms in Parallel Cloud-Edge Hybrid Model

Average DelayFigure 12 shows the average delays in the three slices of each tenant when using the MCRA and FLDRA algorithms and the *PH* model. The experimental results are similar to those obtained for the *CO* and *EO* models. Both algorithms satisfy the SLAs of the two tenants in all three slices. However, the MCRA algorithm results in a lower average delay than FLDRA.Resource AvailabilityFigure 13a,d shows the available resources at each node in the eMBB slice of the two tenants when using the MCRA and FLDRA algorithms and the *PH* model. The standard deviations of the available resources at each node are around 0.01–0.05 and 0.1–0.15 for the two algorithms, respectively. In other words, MCRA once again results in a more uniform resource utilization than FLDRA for both tenants. Additionally, although T1 requires fewer resources in the eMBB slice than T2 and has higher latency constraints, it has a longer survival time (5000 ms) than T2 (1000 ms) and hence consumes significantly more resources. Moreover, when comparing MCRA and FLDRA, due to FLDRA prioritizes the selection of nodes which achieve the greatest reduction in the E2E delay, and among the *EO* model and *CO* model, the former has a higher latency than the latter. Thus, in the *PH* model, FLDRA leans towards assigning requests to the edge rather than the C, thereby increasing the resource usage at the edge and RN nodes. Furthermore, if the latency constraints cannot be satisfied, the FLDRA allocation mechanism distributes requests equally. However, due to the relatively scarce resources available at the E compared to the C, the E quickly becomes overwhelmed, leading to a limitation of both C and CN usage.Figure 13b,e shows the resource availability at each node in the URLLC slice of the *PH* model. The standard deviation of the resource availability under the MCRA algorithm is around 0.02, while that under FLDRA is 0.15–0.2. In other words, MCRA results in a more uniform distribution of the resources than FLDRA for both tenants. As with the *CO* and *EO* models, T1 requires more resources in the URLLC slice than T2. As a result, even though T2 has lower latency constraints than T1, its impact on the resource usage is not as significant as that for T1. In the comparison of MCRA and FLDRA, FLDRA prioritizes the allocation of requests to E, leading to a significant increased in the resource usage at the edge and RN nodes. Furthermore, when the latency constraints cannot be satisfied, FLDRA adopts a more balanced allocation of the requests, resulting in only a limited usage of the C and CN resources.Figure 13c,f shows the resource availability at each node in the mMTC slices of the two tenants in the *PH* model when using the two different algorithms. The standard deviations of the resource availability across the different nodes are around 0.015 for the MCRA algorithm and 0.1–0.2 for the FLDRA algorithm. Thus, as with the *EO* model and *CO* model, the MCRA algorithm achieves a more uniform resource utilization than the FLDRA algorithm for both tenants. In the mMTC slice, T1 requires only a few resources to process its service requests. However, its long survival time results in a significant reduction in the resource availability as the request arrival rate increases. Conversely, T2 requires a greater amount of resources to satisfy its service requests, and thus blocking occurs when the resource availability reduces to around 25%. Finally, as discussed above, the tendency of the FLDRA algorithm to assign service requests to E processing increases the use of the E and RN resources. Moreover, the adoption of a more balanced allocation when the latency constraints cannot be satisfied results in a limited usage of the C and CN resources.Blocking RateFigure 14 shows the blocking rates of the MCRA and FLDRA algorithms in the three slices of the two tenants in the *PH* model. As with the previous models, the blocking rate of the FLDRA algorithm is higher than that of the MCRA algorithm in all six slices.

#### 6.2.4. Comparison of MCRA and UFLOP Algorithms in Cloud-Only Model

Average DelayFigure 15 shows the average delays of the MCRA and UFLOP algorithms in the three slices of the two tenants in the *CO* model. The horizontal axis represents the arrival rate of the requests for the entire system, and the vertical axis represents the average delay. It is seen that both algorithms satisfy the SLA for the E2E delay of each tenant in every slice. However, for all six slices, the average delay of the MCRA algorithm is lower than that of UFLOP.Resource AvailabilityFigure 16 shows the resource availability at each node of the six slices under the MCRA and UFLOP algorithms, respectively. For both algorithms, the standard deviation of the resource availability across the different nodes is around 0.03. In other words, both algorithms yield a relatively uniform distribution of the resource usage across the nodes in the *CO* model. For the UFLOP algorithm, T1 and T2 both suffer blocking in all three slices when the request arrival rate exceeds 30 (see Figure 17c). This observation suggests that the available resources of each node have reached a state of minimum capacity, rendering them unable to accommodate subsequent requests.Blocking RateFigure 17 shows the blocking rates at each node of the six slices of the two tenants under the MCRA and UFLOP algorithms and the *CO* model. The blocking rates of the UFLOP algorithm are higher than those of MCRA in all six slices. In the UFLOP algorithm, all of the slices experience blocking after the arrival of the first few service requests since, because all requests in the same slice do not consider current usage for dynamic adjustment, namely given a fixed resource. By contrast, the MCRA algorithm shares slice resources among all requests from the same tenant and dynamically adjusts resource allocation for each request based on current usage, thus satisfying its E2E constraints.

#### 6.2.5. Comparison of MCRA and UFLOP Algorithms in Edge-Only Model

Average DelayFigure 18 compares the average delays of the MCRA and UFLOP algorithms in the three slices of each tenant in the *EO* model. As with the case of the *CO* model, both algorithms satisfy the SLA requirements of the two tenants in every slice. However, the average delay under the MCRA algorithm is lower than that under the UFLOP algorithm.Resource AvailabilityFigure 19 presents the resource availability at each node in three slices of the *EO* model using MCRA and UFLOP. For both MCRA and UFLOP, the standard deviation of the resource availability at each node across six different slices is approximately 0.02, indicating a relatively uniform utilization of resources. However, when the arrival rate reaches 30, the UFLOP algorithm experiences blocking in both T1 and T2 (see Figure 20c), which leads to the inability to accommodate the next request. This trend is supported by the halt in the decline of remaining resource availability.Blocking RateFigure 20 compares the blocking rates of the MCRA and UFLOP algorithms in the three slices of each tenant under the *EO* model. It is evident that UFLOP has a higher blocking rate than MCRA across all six slices since, as discussed above for the *CO* model, the UFLOP algorithm can only accommodate a small number of requests before blocking occurs due to the fixed manner in which the resources are allocated within the slice.

#### 6.2.6. Performance of MCRA and UFLOP Algorithms in Parallel Cloud-Edge Hybrid Model

Average DelayFigure 21 shows the average delays in the three slices of the tenants when using the MCRA and UFLOP algorithms in the *PH* model. Both algorithms satisfy the SLA requirements of the two tenants in all six slices. However, the average delay under the MCRA algorithm is lower than that under UFLOP.Resource AvailabilityFigure 22 shows the resource availability at each node in the three slices of each tenant given the use of the MCRA and UFLOP algorithms and the *PH* model. For each slice, and both tenants, the standard deviation of the resource availability across the nodes is around 0.02–0.05. Thus, both algorithms result in a relatively balanced resource utilization within each slice. For the UFLOP algorithm, blocking occurs for both tenants when the arrival rate reaches 30 (see Figure 23), which is unable to accommodate the next request.Blocking RateFigure 23 shows the blocking rates of the MCRA and UFLOP algorithms in the different slices of the two tenants under the *PH* model. For all six slices, the blocking rate under UFLOP is higher than that under MCRA. This result is consistent with that obtained under the *CO* and *EO* models and confirms that the UFLOP algorithm accommodates significantly fewer requests than MCRA, thus leading to an early blocking of new service requests.

#### 6.2.7. Overall Performance Comparison

Comparison of Three Algorithms in Same ModelFigure 24 shows the blocking rates of the MCRA, FLDRA, and UFLOP algorithms for the three slices of each tenant in the *CO*, *EO*, and *PH* models. Overall, the results show that, for each model, the MCRA algorithm accommodates the highest request arrival rate, followed by the FLDRA algorithm and then the UFLOP algorithm. As described in Section 6.2.1, Section 6.2.2 and Section 6.2.3, MCRA outperforms FLDRA since it calculates weights based on the current resource usage of each node, and selects nodes with more remaining resources for allocation, resulting in a more evenly distributed amount of remaining resources. On the other hand, FLDRA adjusts the node with the most significant delay reduction, leading to a similar number of adjustments for each node, without prioritizing nodes with more remaining resources. In summary, MCRA can achieve a higher arrival rate without causing blocking, while FLDRA has a lower rate. Furthermore, in Section 6.2.4, Section 6.2.5 and Section 6.2.6, the UFLOP algorithm is prone to rapid blocking of all slices. This is because UFLOP does not dynamically adjust the allocation of resources based on the current node usage. As a result, the resources within the slice are quickly consumed, and new-arriving requests are blocked.Comparison of Same Algorithm on Different ModelsFigure 24c,f,i shows the blocking rates in the different slices when using the same algorithm (UFLOP) in the different models. It is observed that the *CO* model results in the earliest occurrence of blocking, followed by the *EO* model, and the *PH* model. The relatively poorer performance of the *CO* model stems for its use of a large number of nodes, i.e., C, CU, CD, RU and RD. By contrast, the *EO* model uses just three nodes, i.e., E, RU and RD. Therefore, given the same latency constraint, the *EO* model consumes fewer resources for each request and therefore experiences later blocking. The *PH* model has the advantage of cloud-side diversion, which is not available in the *EO* model. This allows the service requests to be processed either in the C or at the E depending on the current network situation, and hence the blocking rate is reduced.Comparison of Three Algorithms Used in Different ModelsFigure 25 shows the blocking rates in each slice of the two tenants when using the three resource allocation algorithms and different network structure models. For each slice, the UFLOP algorithm results in the earliest occurrence of blocking, followed by FLDRA and finally MCRA. Furthermore, for a given resource allocation algorithm, the *CO* model results in the quickest blocking speed, followed by the *EO* model, and then the *PH* model.

## 7. Conclusions

In previous related work, most One-tier architectures ignored the architecture of cloud and edge networks and only considered computing resources, neglecting communication resources. In the related work on Two-tier Architecture, most did not consider the latency constraints specified in the SLA of the slice, with the exception of Chien et al.’s relatively comprehensive study. However, Chien et al.’s method have three limitations: (1) it does not consider the core network, (2) it does not account for the lack of queueing models, including improper consideration of the tandem queues and the handling of each service request separately, and (3) it does not propose an optimization strategy for service triage. To address these issues, we propose two algorithms (MCRA and FLDRA) and three models (*CO*, *EO*, and *PH*). First, we consider the core network and use queuing models to establish a network architecture that aligns with reality. Second, we allocate separate resources for each service request and dynamically allocate the required resources for each request. Third, we propose the MCRA and FLDRA dynamic resource allocation algorithms for 5G two-tier multi-user network slicing, which are coordinated with the *CO*, *EO*, and *PH* network structures. We also consider the dynamic resource allocation depending on the real-time resource usage conditions of the network and optimize task offloading. Finally, our goal is to maximize resource utilization, minimize E2E delay, slice blocking rate, and improve overall system performance while satisfying QoS requirements.

Specifically, this study has considered the problem of resource allocation within the eMBB, URLLC and mMTC slices of different tenants in a two-tier architecture. Two algorithms (MCRA and FLDRA) have been proposed for performing dynamic resource allocation depending on the real-time resource usage conditions of the network. In performing resource allocation, MCRA favors the nodes with the minimum cost, while FLDRA favors the nodes which minimize the E2E delay. The overall goal of both algorithms is to optimize the resource allocation and maximize the service capacity of each network slice subject to the constraint of satisfying the service quality requirements of the multiple tenants.

Three network structure models have been considered: *CO*, *EO*, and *PH*. The performance of the MCRA and FLDRA algorithms has been evaluated and compared for each model. The simulation results have shown that both algorithms satisfy the SLAs of the tenants in all three slices, irrespective of the network model used. However, of the two algorithms, MCRA achieves the best resource availability as it allocates the resources based on cost. Furthermore, of the three network structure models, the *PH* model can accommodate the highest service request arrival rate due to its ability to support cloud-side diversion.

It has been shown that the UFLOP algorithm [20] also satisfies the specified latency requirements of the tenant in each slice. However, due to the lack of dynamic resource allocation, in the early stages, the remaining resource availability under the UFLOP algorithm is low, leading to a higher blocking rate. As a result, the remaining resource availability under the MCRA and FLDRA algorithms is higher than that under the UFLOP algorithm.

In summary, the experimental results have shown that the MCRA and FLDRA algorithms outperform the UFLOP algorithm proposed by Chien et al. [20]. Additionally, MCRA provides better network performance, as it calculates weights based on the current resource usage of each node, resulting in a more evenly distributed residual resource availability. As for FLDRA, it adjusts the node with the most significant decrease in delay, without prioritizing a node with more remaining resources. Overall, MCRA can achieve a higher arrival rate without causing blocking. Finally, based on same blocking rate, the *PH* model can accommodate a higher arrival rate, meaning that the *PH* model delays the onset of blocking as the arrival rate increases.

In this paper, we made several assumptions on traffic patterns, such as the Poisson arrival process and exponential service time. Thus, the results presented in this paper may not be valid to general traffic patterns. The allocation and offloading of computing resources in multi-layer and multi-tenant MEC is a critical issue in 5G networks. In future work, we propose to investigate the extension of vertical offloading to horizontal offloading to reduce blocking rate and explore the diverse slice types specified in 6G. 

## Figures and Tables

**Figure 1 sensors-23-04698-f001:**
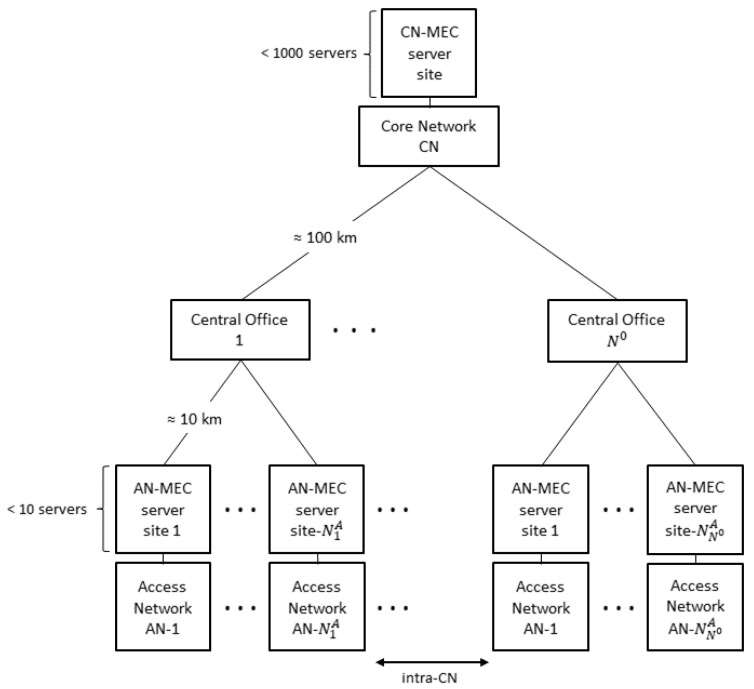
Two-tier network architecture.

**Figure 2 sensors-23-04698-f002:**
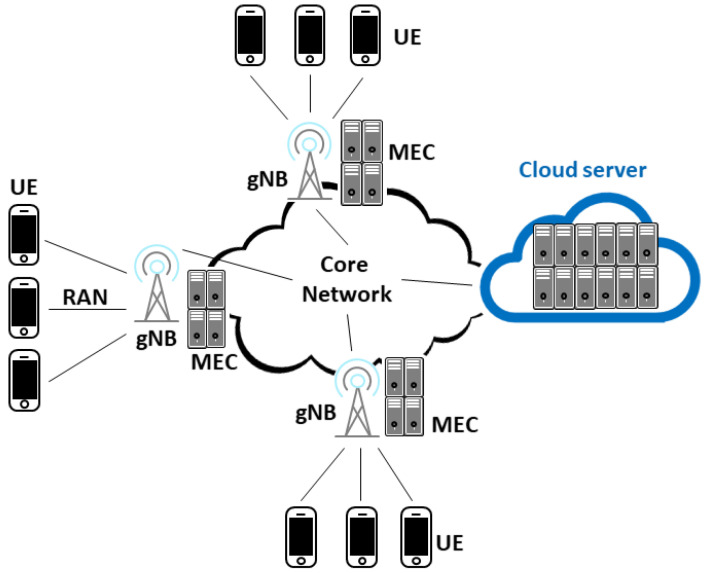
Simplified architecture of two-tier networks.

**Figure 3 sensors-23-04698-f003:**
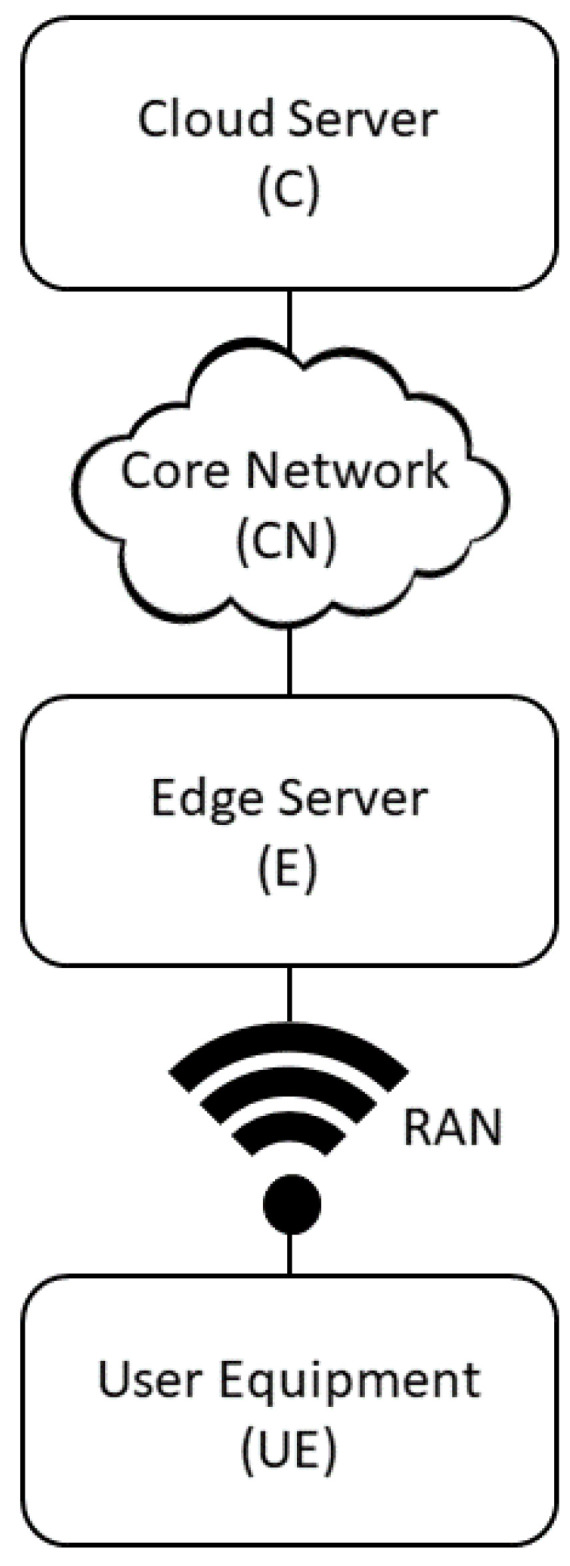
Packet flow architecture in a two-tier network.

**Figure 4 sensors-23-04698-f004:**
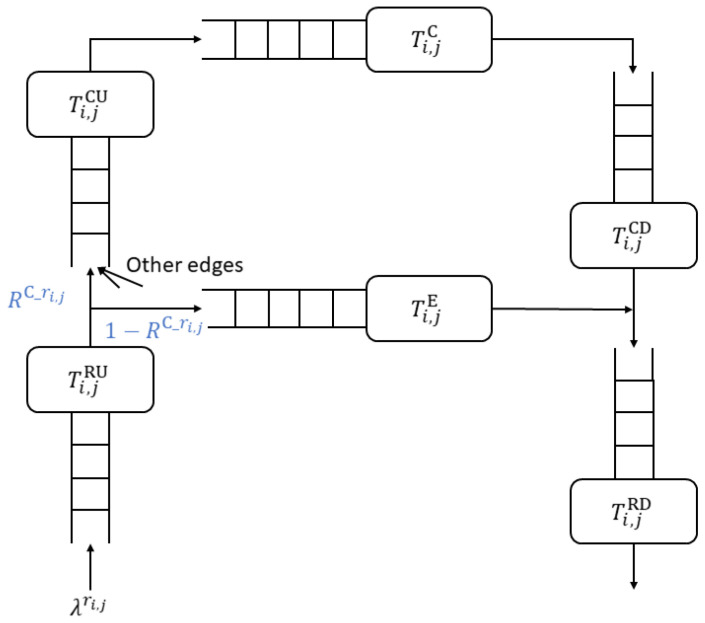
Queueing network in proposed two-tier architecture.

**Figure 5 sensors-23-04698-f005:**
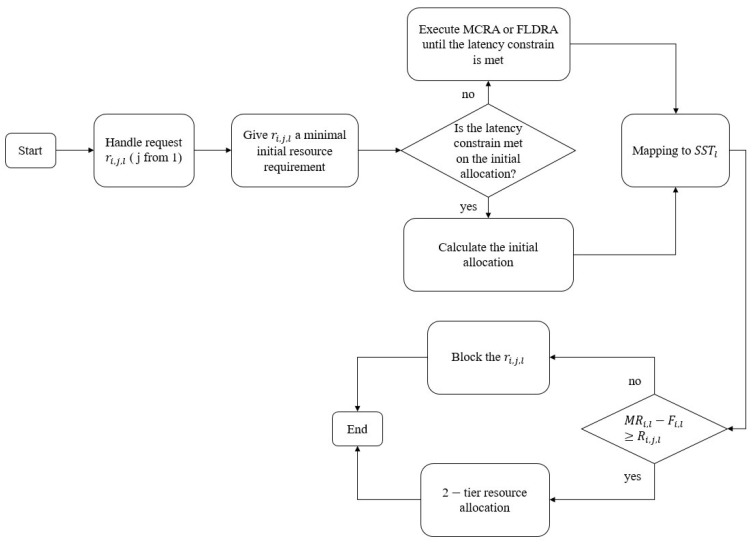
Flowchart of resource allocation method in 5G network slicing.

**Figure 6 sensors-23-04698-f006:**
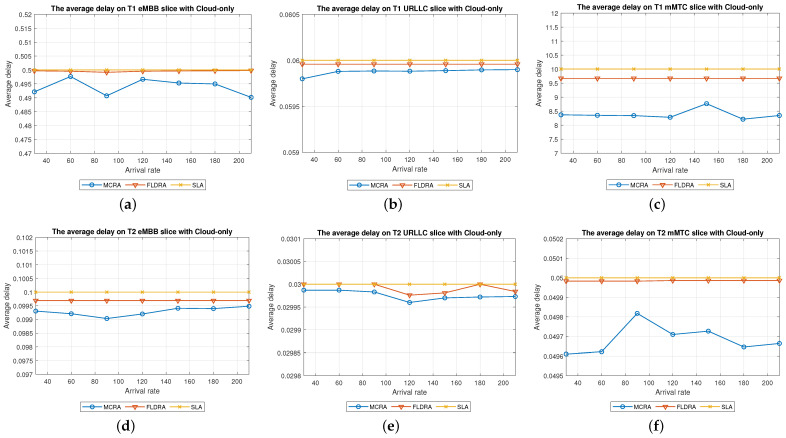
Comparison of average delays in different slices of each tenant under MCRA and FLDRA algorithms and Cloud-only (*CO*) model. (**a**) T1 eMBB. (**b**) T1 URLLC. (**c**) T1 mMTC. (**d**) T2 eMBB. (**e**) T2 URLLC. (**f**) T2 mMTC.

**Figure 7 sensors-23-04698-f007:**
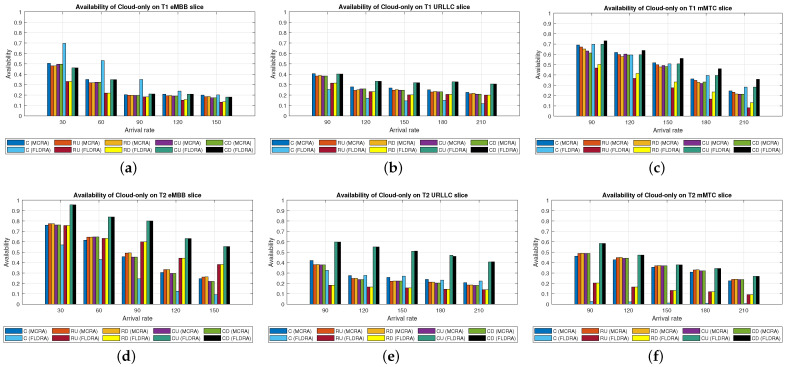
Comparison of resource availability in different slices of each tenant under MCRA and FLDRA algorithms and *CO* model. (**a**) T1 eMBB. (**b**) T1 URLLC. (**c**) T1 mMTC. (**d**) T2 eMBB. (**e**) T2 URLLC. (**f**) T2 mMTC.

**Figure 8 sensors-23-04698-f008:**
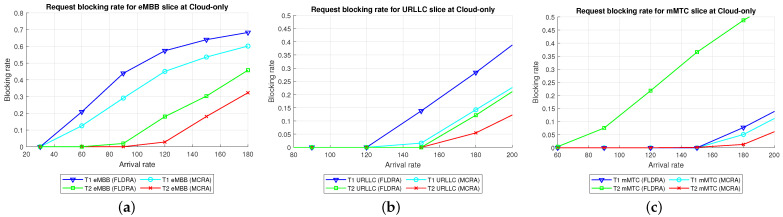
Comparison of blocking rates in different slices of each tenant under MCRA and FLDRA algorithms and *CO* model. (**a**) eMBB slice at *CO*. (**b**) URLLC slice at *CO*. (**c**) mMTC slice at *CO*.

**Figure 9 sensors-23-04698-f009:**
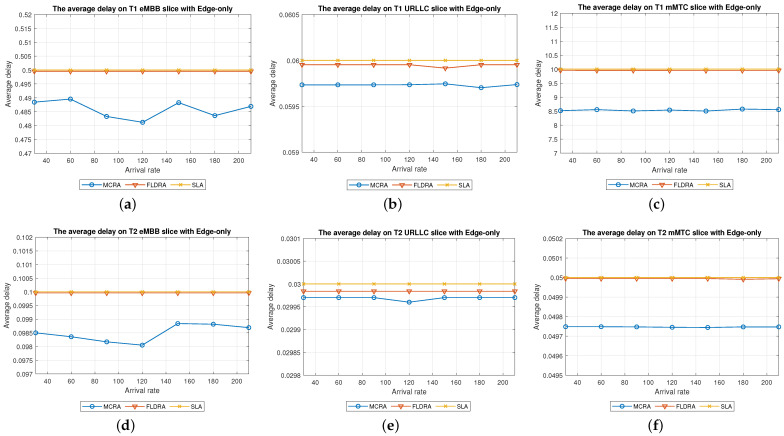
Comparison of average delays in different slices of each tenant under MCRA and FLDRA algorithms and Edge-only (*EO*) model. (**a**) T1 eMBB. (**b**) T1 URLLC. (**c**) T1 mMTC. (**d**) T2 eMBB. (**e**) T2 URLLC. (**f**) T2 mMTC.

**Figure 10 sensors-23-04698-f010:**
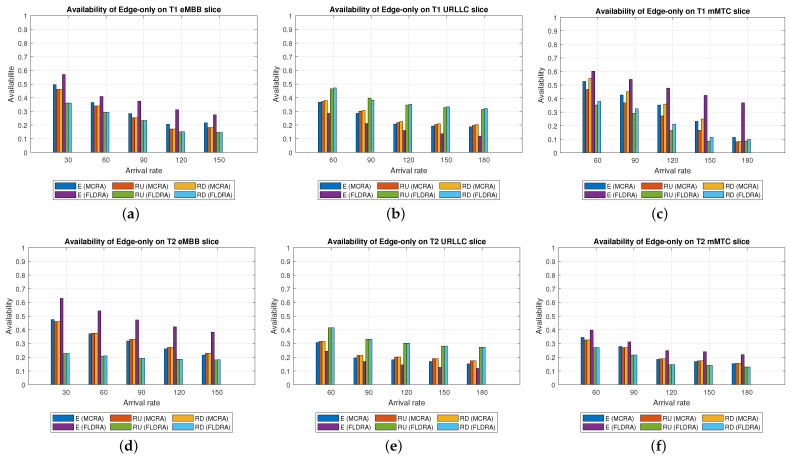
Comparison of resource availability in different slices of each tenant under MCRA and FLDRA algorithms and *EO* model. (**a**) T1 eMBB. (**b**) T1 URLLC. (**c**) T1 mMTC. (**d**) T2 eMBB. (**e**) T2 URLLC. (**f**) T2 mMTC.

**Figure 11 sensors-23-04698-f011:**
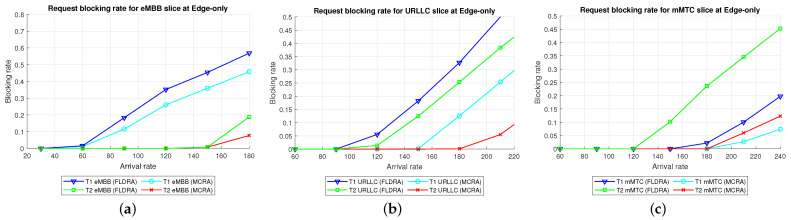
Comparison of blocking rates in different slices of each tenant under MCRA and FLDRA algorithms and *EO* model. (**a**) eMBB slice at *EO*. (**b**) URLLC slice at *EO*. (**c**) mMTC slice at *EO*.

**Figure 12 sensors-23-04698-f012:**
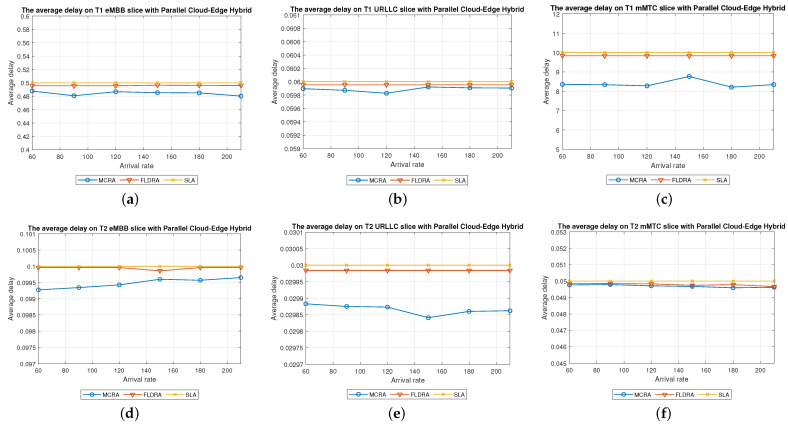
Comparison of average delays in different slices of each tenant under MCRA and FLDRA algorithms and Parallel Cloud-Edge Hybrid (*PH*) model. (**a**) T1 eMBB. (**b**) T1 URLLC. (**c**) T1 mMTC. (**d**) T2 eMBB. (**e**) T2 URLLC. (**f**) T2 mMTC.

**Figure 13 sensors-23-04698-f013:**
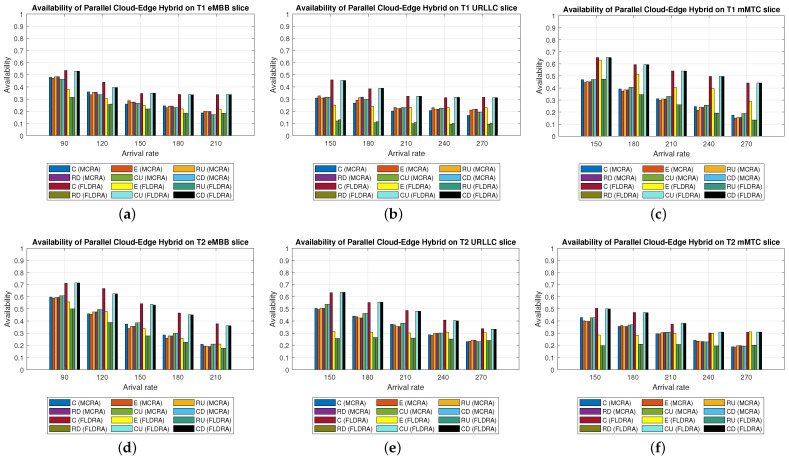
Comparison of resource availability in different slices of each tenant under MCRA and FLDRA algorithms and *PH* model. (**a**) T1 eMBB. (**b**) T1 URLLC. (**c**) T1 mMTC. (**d**) T2 eMBB. (**e**) T2 URLLC. (**f**) T2 mMTC.

**Figure 14 sensors-23-04698-f014:**
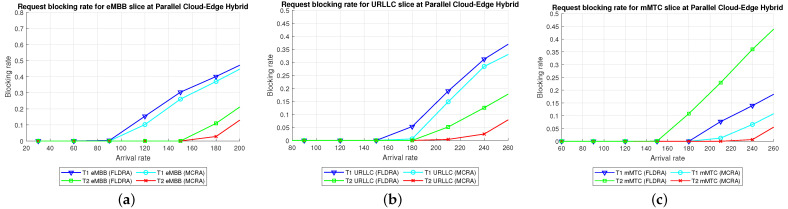
Comparison of blocking rates in different slices of each tenant under MCRA and FLDRA algorithms and *PH* model. (**a**) eMBB slice at *PH*. (**b**) URLLC slice at *PH*. (**c**) mMTC slice at *PH*.

**Figure 15 sensors-23-04698-f015:**
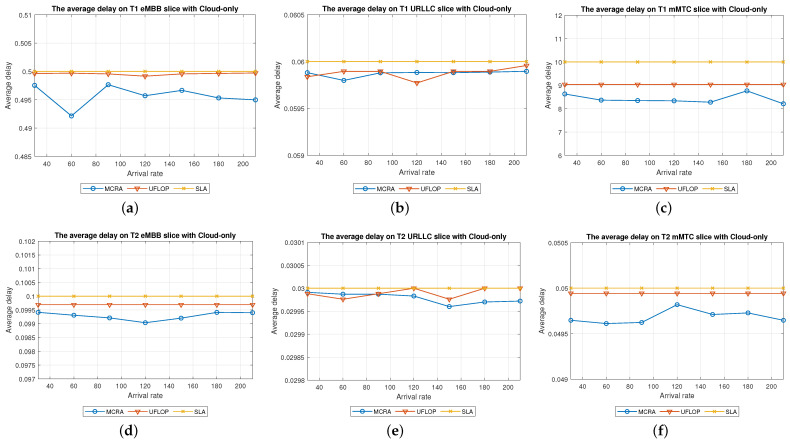
Comparison of average delay in different slices of each tenant under MCRA and UFLOP algorithms and *CO* model. (**a**) T1 eMBB. (**b**) T1 URLLC. (**c**) T1 mMTC. (**d**) T2 eMBB. (**e**) T2 URLLC. (**f**) T2 mMTC.

**Figure 16 sensors-23-04698-f016:**
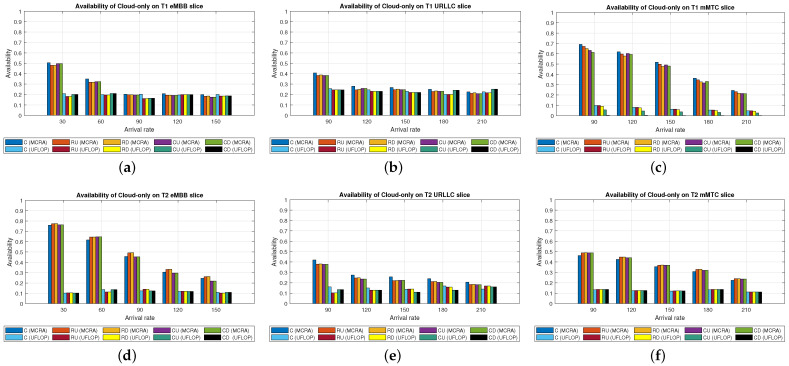
Comparison of resource availability in different slices of each tenant under MCRA and UFLOP algorithms and *CO* model. (**a**) T1 eMBB. (**b**) T1 URLLC. (**c**) T1 mMTC. (**d**) T2 eMBB. (**e**) T2 URLLC. (**f**) T2 mMTC.

**Figure 17 sensors-23-04698-f017:**
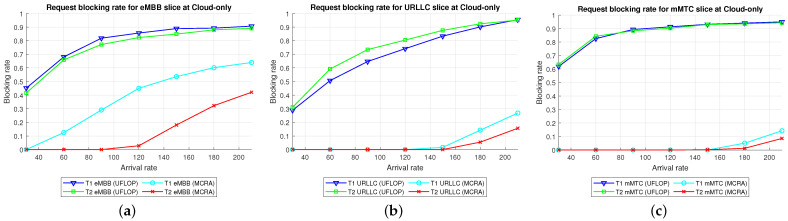
Comparison of blocking rates in different slices of each tenant under MCRA and UFLOP algorithms and *CO* model. (**a**) eMBB slice at *CO*. (**b**) URLLC slice at *CO*. (**c**) mMTC slice at *CO*.

**Figure 18 sensors-23-04698-f018:**
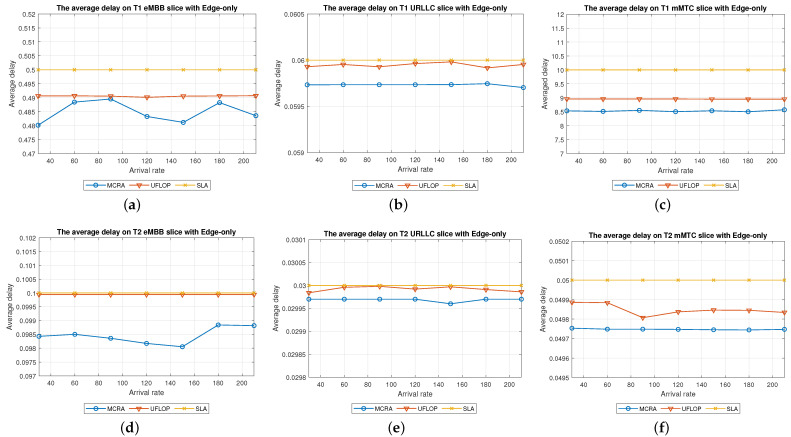
Comparison of average delays in different slices of each tenant under MCRA and UFLOP algorithms and *EO* model. (**a**) T1 eMBB. (**b**) T1 URLLC. (**c**) T1 mMTC. (**d**) T2 eMBB. (**e**) T2 URLLC. (**f**) T2 mMTC.

**Figure 19 sensors-23-04698-f019:**
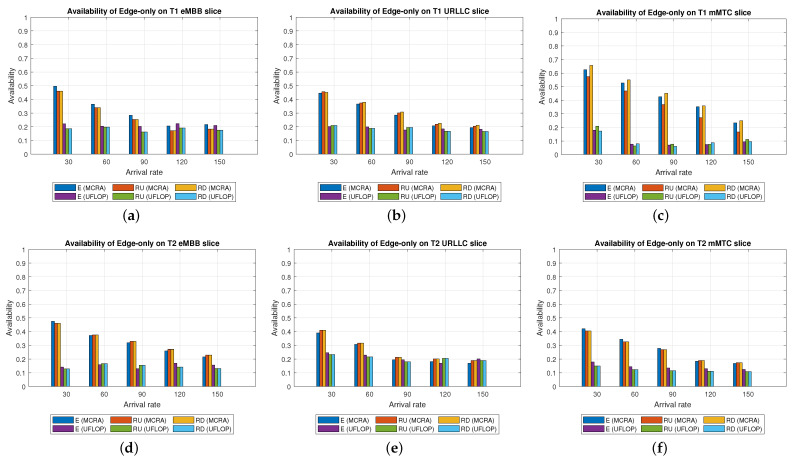
Comparison of resource availability in different slices of each tenant under MCRA and UFLOP algorithms and *EO* model. (**a**) T1 eMBB. (**b**) T1 URLLC. (**c**) T1 mMTC. (**d**) T2 eMBB. (**e**) T2 URLLC. (**f**) T2 mMTC.

**Figure 20 sensors-23-04698-f020:**
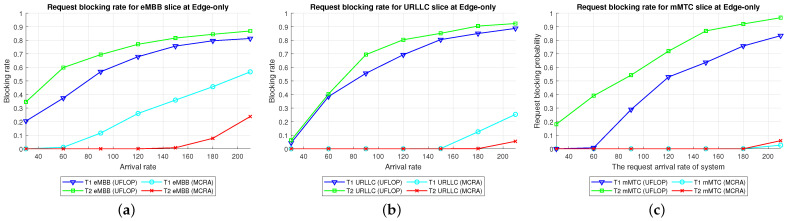
Comparison of blocking rates in different slices of each tenant under MCRA and UFLOP algorithms and *EO* model. (**a**) eMBB slice at *EO*. (**b**) URLLC slice at *EO*. (**c**) mMTC slice at *EO*.

**Figure 21 sensors-23-04698-f021:**
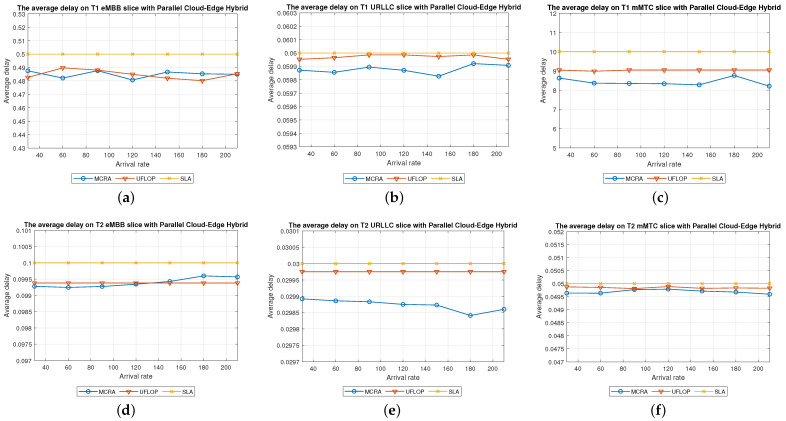
Comparison of average delay in different slices of each tenant under MCRA and UFLOP algorithms and *PH*. (**a**) T1 eMBB. (**b**) T1 URLLC. (**c**) T1 mMTC. (**d**) T2 eMBB. (**e**) T2 URLLC. (**f**) T2 mMTC.

**Figure 22 sensors-23-04698-f022:**
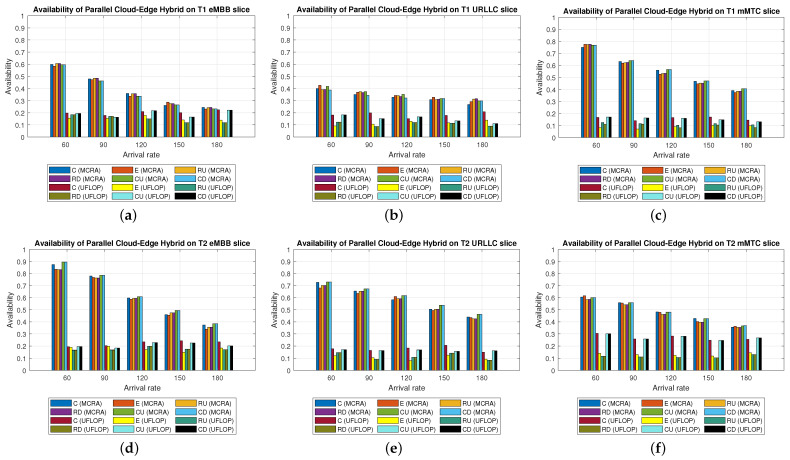
Comparison of resource availability in different slices of each tenant under MCRA and UFLOP algorithms and *PH* model. (**a**) T1 eMBB. (**b**) T1 URLLC. (**c**) T1 mMTC. (**d**) T2 eMBB. (**e**) T2 URLLC. (**f**) T2 mMTC.

**Figure 23 sensors-23-04698-f023:**
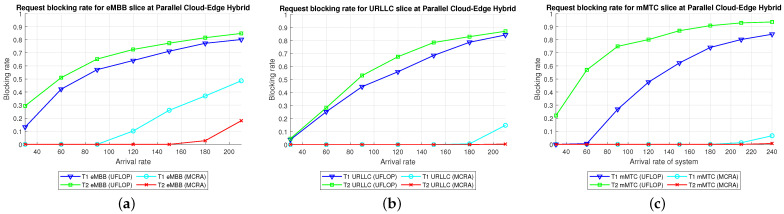
Comparison of blocking rates in different slices of each tenant under MCRA and UFLOP algorithms and *PH* model. (**a**) eMBB slice at *PH*. (**b**) URLLC slice at *PH*. (**c**) mMTC slice at *PH*.

**Figure 24 sensors-23-04698-f024:**
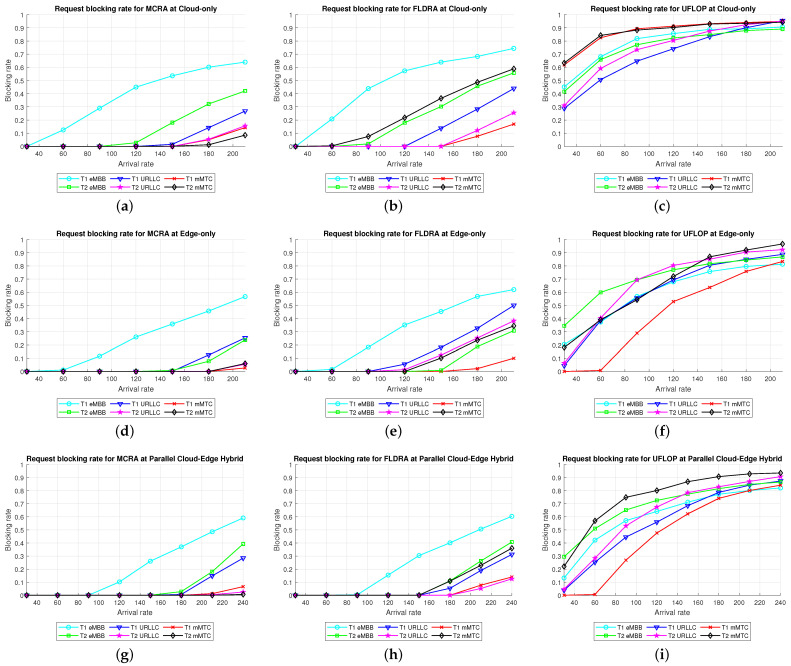
Comparison of blocking rate of different algorithms in slices of three tenants under different network structure models. (**a**) The MCRA algorithm in the *CO* model. (**b**) The FLDRA algorithm in the *CO* model. (**c**) The UFLOP algorithm in the *CO* model. (**d**) The MCRA algorithm in the *EO* model. (**e**) The FLDRA algorithm in the *EO* model. (**f**) The UFLOP algorithm in the *EO* model. (**g**) The MCRA algorithm in the *PH* model. (**h**) The FLDRA algorithm in the *PH* model. (**i**) The UFLOP algorithm in the *PH* model.

**Figure 25 sensors-23-04698-f025:**
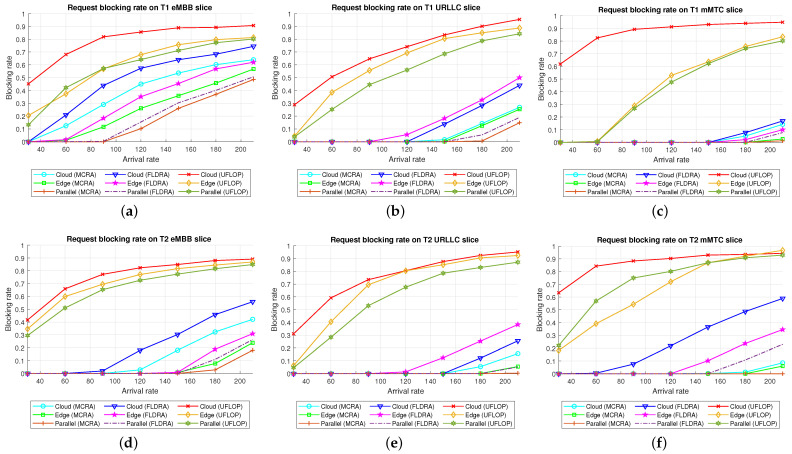
Comparison of blocking rates in each slice given different algorithms and network structure models. (**a**) T1 eMBB. (**b**) T1 URLLC. (**c**) T1 mMTC. (**d**) T2 eMBB. (**e**) T2 URLLC. (**f**) T2 mMTC.

**Table 1 sensors-23-04698-t001:** Comparison of Related Works.

Lit.	Proposed	Network/Server Slicing (RAN/E/CN/C)	Tiers	Dynamic Allocation	Latency Constraint
[15]	Concept Model Algorithm	Network -/-/CN/-	1	No	No
[16]	Concept	Network RAN/-/CN/-			
[17]	Architecture	Both -/-/N/C			
[18]	Architecture Algorithm	Both -/E/-/C			
[19]	Architecture Algorithm Workflow	Both RAN/E/-/-		Yes	Yes
[21]	Architecture Algorithm	Both -/E/-/C	2	No	No
[22]	Architecture	Both RAN/E/-/C			
[23]	Architecture Algorithm	Both -/E/-/C			
[24]	Architecture Algorithm	Both RAN/E/CN/C		No	Yes
[20]	Architecture Algorithm Workflow	Both RAN/E/CN/C			
Proposed	Architecture Algorithm Workflow	Both RAN/E/CN/C		Yes	Yes

**Table 2 sensors-23-04698-t002:** Notations.

	Symbol	Description
Entity	C	Cloud Server
	E	Edge server
	UE	User Equipment
	CN	Core Network
	RN	RAN
	*L*	Number of service types (slices) supported (e.g., eMBB, URLLC, mMTC)
Tenant	*T*, $T_{i}$	$T_{i}$ is the *i*-th tenant, *T* = $T_{i}$
	$T_{i,j}^{\mathrm{XU}}$, $T_{i,j}^{\mathrm{XD}}$	The *j*-th uplink or downlink request of the *i*-th tenant on different node (X)
	$S_{i,l}$	The *l*-th slice of the *i*-th tenant
	$SLA_{i,l}$	SLA of $S_{i,l}$
	$F_{i,l}^{\mathrm{RU}}$, $F_{i,l}^{\mathrm{RD}}$	Current RN uplink and downlink resources allocated to $S_{i,l}$
	$F_{i,l}^{\mathrm{CU}}$, $F_{i,l}^{\mathrm{CD}}$	Current CN uplink and downlink resources allocated to $S_{i,l}$
	$F_{i,l}^{C}$, $F_{i,l}^{E}$	Current C and E computing resources allocated to $S_{i,l}$
	${\tilde{\lambda }}_{i,l}$	Current C and E computing resources allocated to $S_{i,l}$
	$A_{i,l}^{\mathrm{RU}}$, $A_{i,l}^{\mathrm{RD}}$	Availability of RN uplink and downlink resources of $S_{i,l}$. $A_{i,l}^{\mathrm{RU}}=1-\frac{F_{i,l}^{\mathrm{RU}}}{MR_{i,l}^{\mathrm{RU}}}$, $A_{i,l}^{\mathrm{RU}}=1-\frac{F_{i,l}^{\mathrm{RD}}}{MR_{i,l}^{\mathrm{RD}}}$
	$A_{i,l}^{\mathrm{CU}}$, $A_{i,l}^{\mathrm{CD}}$	Availability of CN uplink and downlink resources of $S_{i,l}$. $A_{i,l}^{\mathrm{CU}}=1-\frac{F_{i,l}^{\mathrm{CU}}}{MR_{i,l}^{\mathrm{CU}}}$, $A_{i,l}^{\mathrm{CU}}=1-\frac{F_{i,l}^{\mathrm{CD}}}{MR_{i,l}^{\mathrm{CD}}}$
	$A_{i,l}^{C}$, $A_{i,l}^{E}$	Availability of C and E computing resources of $S_{i,l}$. $A_{i,l}^{C}=1-\frac{F_{i,l}^{C}}{MR_{i,l}^{C}}$, $A_{i,l}^{E}=1-\frac{F_{i,l}^{E}}{MR_{i,l}^{E}}$
	$SA_{i,l}$	Slice resource availability of $S_{i,l}$. $SA_{i,l}$ = $A_{i,l}^{\mathrm{RU}}+A_{i,l}^{\mathrm{RD}}+A_{i,l}^{\mathrm{CU}}+A_{i,l}^{\mathrm{CD}}+A_{i,l}^{C}+A_{i,l}^{E}$
SLA	$MR_{i,l}^{\mathrm{RU}}$, $MR_{i,l}^{\mathrm{RD}}$	Guaranteed (maximum) RN uplink and downlink resources specified by $SLA_{i,l}$
	$MR_{i,l}^{\mathrm{CU}}$, $MR_{i,l}^{\mathrm{CD}}$	Guaranteed (maximum) CN uplink and downlink resources specified by $SLA_{i,l}$
	$MR_{i,l}^{C}$, $MR_{i,l}^{E}$	Guaranteed (maximum) C and E computing resources specified by $SLA_{i,l}$
	$D_{i,l}$	The end-to-end (E2E) delay requirement of slice $S_{i,l}$
Service Deployment Request	$\Lambda_{i,l}$	Request arrival rate of slice $S_{i,l}$
	$M_{i,l}$	Request departure rate of slice $S_{i,l}$
	$r_{i,j,l}$	The *j*-th request of slice $S_{i,l}$
	$\lambda_{i,j,l}$	The traffic arrival rate of $r_{i,j,l}$
	$\mu_{i,j,l}^{C}$, $\mu_{i,j,l}^{E}$	The computing resource demand of $r_{i,j,l}$
	$\mu_{i,j,l}^{\mathrm{NU}}$, $\mu_{i,j,l}^{\mathrm{ND}}$	The network (communication) resource demand of $r_{i,j,l}$
	$R_{i,j,l}^{\mathrm{RU}}$, $R_{i,j,l}^{\mathrm{RD}}$	RN uplink and downlink resources allocated to $r_{i,j,l}$
	$R_{i,j,l}^{\mathrm{CU}}$, $R_{i,j,l}^{\mathrm{CD}}$	CN uplink and downlink resources allocated to $r_{i,j,l}$
	$R_{i,j,l}^{C}$, $R_{i,j,l}^{E}$	Cloud and edge computing resources allocated to $r_{i,j,l}$
Delay	$D_{i,j,l}^{C}$	The estimated computing delay of C of $r_{i,j,l}$
	$D_{i,j,l}^{E}$	The estimated computing delay of E of $r_{i,j,l}$
	$D_{i,j,l}^{\mathrm{RU}}$,$D_{i,j,l}^{\mathrm{RD}}$	The estimated network delay of uplink and downlink of RN of $r_{i,j,l}$
	$D_{i,j,l}^{\mathrm{CU}}$,$D_{i,j,l}^{\mathrm{CD}}$	The estimated network delay of uplink and downlink of CN of $r_{i,j,l}$
	$D_{i,j,l}^{w}$	The estimated network delay of different place *m* of $r_{i,j,l}$ *w* = Net_C,Net_E,Comp_C,Comp_E,E2E_C,E2E_E, and P_m, where m∈{Cloud-only (CO), Edge-only (EO)}
Unit Resource Comparison	$\gamma^{X}$	The cost of allocating one unit of resource at node X, where X is C, E, RU, RD, CU, CD
	*∂*	The parameter of the resource reservation cost function (default value is 100)
	α	The parameter for setting the smallest resource unit
	$Cost_{i,j,l}^{PHC},Cost_{i,j,l}^{PHE}$	The total E2E cost of $r_{i,j,l}$ using *CO*/*EO* in the Parallel Cloud-Edge Hybrid (*PH*) model
	$D_{i,j,l}^{PHC},D_{i,j,l}^{PHE}$	The total E2E delay of $r_{i,j,l}$ using *CO*/*EO* in the *PH* model
	$D_{unit}^{X}$	The reduction of delay resulting from resource allocation at node X

**Table 3 sensors-23-04698-t003:** Parameter settings.

	Tenant 1			Tenant 2		
	eMBB	URLLC	mMTC	eMBB	URLLC	mMTC
$\mu^{\mathrm{NU}}$ (Uplink resource demand, unit: Mbps)	2.5	20	0.14	4	2	0.1
$\mu^{\mathrm{ND}}$ (Downlink resource demand, unit: Mbps)	5	1	0.125	8	0.2	0.05
$D_{i,l}$ (Latency constraints, unit: ms)	500	60	10,000	100	30	50
Survival time (ms)	5000	100	10,000	1000	100	500
$M_{i,l}$ (Departure rate)	0.2	10	0.1	1	10	2
$\lambda_{i,j,l}$ (Traffic arrival rate, unit: packet/request)	10	1	1	10	5	5

## Data Availability

Not applicable.

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
