# Peer review of "Dynamic Resource Allocation for Network Slicing with Multi-Tenants in 5G Two-Tier Networks"

_sensors, 2023, doi:10.3390/s23104698_

Round 1
Reviewer 1 Report
The paper proposes two heuristic algorithms, namely Minimum Cost Resource Allocation (MCRA) and Fast Latency Decrease Resource Allocation (FLDRA), to perform dynamic path routing and resource allocation for multi-tenant network slices in a two-tier architecture. The topic of research is interesting and represents a highly studied area of research in recent years.
In the Introduction, the most important notions used in the paper are explained. These are: Fifth Generation (5G) network, Software Defined Network (SDN), Network Function Virtualization (NFV), 3rd Generation Partnership Project (3GPP), Network slicing etc. The Introduction is well-written and all references are related to the terms. The only exception is the QoS notion. No reference are made to the notion of Quality of Service (QoS)-aware network slicing mechanism. A good source to this notion is the paper:
W. -K. Chen, et al., "Optimal Qos-Aware Network Slicing for Service-Oriented Networks with Flexible Routing," ICASSP 2022 - 2022 IEEE International Conference on Acoustics, Speech and Signal Processing (ICASSP), Singapore, Singapore, 2022, pp. 5288-5292, doi: 10.1109/ICASSP43922.2022.9747910.
where a network slicing problem which attempts to map multiple customized virtual network requests (also called services) to a common shared network infrastructure and allocate network resources to meet diverse quality of service (QoS) requirements.
The notion of QoS needs to be introduced in more details. The author should include references to the documents of the International Telecommunication Union for example the QoS regulations (ITU-T Supp. 9 of E.800 Series), The vocabulary for performance, quality of service and quality of experience, etc. Also, in recent years there are studies on QoS and QoE in overall telecommunication systems, addressing important problems related to determining the QoS. For example, an overall normalization approach for determining the QoS in overall telecommunication systems is described in:
S. A. Poryazov, V. S. Andonov, E. T. Saranova, "Overall Model Normalization towards Adequate Prediction and Presentation of QoE in Overall Telecommunication Systems," 2019 14th International Conference on Advanced Technologies, Systems and Services in Telecommunications (TELSIKS), Nis, Serbia, 2019, pp. 360-363.
Another, important related work is:
Manzanares-Lopez, P.; Malgosa-Sanahuja, J.; Muñoz-Gea, J.P. “A Software-Defined Networking Framework to Provide Dynamic QoS Management in IEEE 802.11 Networks”. Sensors 2018, 18, 2247. https://doi.org/10.3390/s18072247.
The above references can be added either in the Introduction or in the Related Work section.
The Related work section consists of two subsection: One-tier architecture and Two-tier architecture. The literature review is extensive and the comparison of the related works is presented in a table (Table 1). The main drawbacks of the related studies are stated.
In view of the drawbacks of the related works, the authors have proposed a system architecture model in Section 3 (two-tier network architecture, packet flow network architecture). The problem is well-defined and all notations are clearly presented in a table (Table 2). I have no critical remarks about the section.
In Section 4 , a Resource Allocation Work Flow model and Latency model are proposed. In the resource allocation procedure each service node is modeled as an M/M/1 queue. The authors should explain the Kendall’s notation as not all of the readers are aware of the theory of queuing systems.
In Section 5, four algorithms are described: for resource allocation for each request in Cloud-only and Edge-only Model; for resource allocation for each request in Parallel Cloud-Edge Hybrid Model; Minimum Cost Resource Allocation Algorithm; Fast Latency Decrease Resource Allocation Algorithm. The algorithms are correctly described and solve the corresponding problems.
The simulations are very carefully carried out and the numerical results show the correctness of the algorithms.
The Conclusions drawn by the authors are supported by the simulation experiments.
Overall, the paper represents a thorough study on path routing and resource allocation for multi-tenant network slices in a two-tier architectures. It is a high quality paper that I recommend to be published once the authors take into account my minor remarks.
The English language is fine.
Reviewer 2 Report
The main contributions and motivations for the present work have to be clearer.
Authors need to confirm that all acronyms are defined before being used
The authors’ examination of the manuscript was not careful enough. For example, when introducing some English abbreviations, their full names are not capitalized,
The contributions of this manuscript need further refinement, and some of them seem insignificant. please describe the features, functions and innovations of the network in more detail.
The main contributions and motivations for the present work have to be clearer.
Authors need to confirm that all acronyms are defined before being used
The authors’ examination of the manuscript was not careful enough. For example, when introducing some English abbreviations, their full names are not capitalized,
The contributions of this manuscript need further refinement, and some of them seem insignificant. please describe the features, functions and innovations of the network in more detail.
Reviewer 3 Report
This paper proposes two heuristic algorithms, namely Minimum Cost Resource Allocation (MCRA) and Fast Latency Decrease Resource Allocation (FLDRA), to perform dynamic path routing and resource allocation for multi-tenant network slices in a two-tier architecture. Overall the proposed algorithms are proven effective and the manuscript is well-written. Some minor suggestions are as follows:
(1) Some abbreviations are defined more than once, e.g., MCRA and FLDRA, which is unnecessary.
(2) Some more recent graph-based methods for network slicing and resource allocation should be added into related work, as listed as follows:
Tam P, Song I, Kang S, et al. Graph Neural Networks for Intelligent Modelling in Network Management and Orchestration: A Survey on Communications[J]. Electronics, 2022, 11(20): 3371.
Jiang W. Graph-based Deep Learning for Communication Networks: A Survey[J]. Computer Communications, 2022, 185:40-54.
Yuan S, Zhang Y, Ma T, et al. Graph convolutional reinforcement learning for resource allocation in hybrid overlay–underlay cognitive radio network with network slicing[J]. IET Communications, 2022.
Shao Y, Li R, Hu B, et al. Graph Attention Network-based Multi-agent Reinforcement Learning for Slicing Resource Management in Dense Cellular Network[J]. IEEE Transactions on Vehicular Technology, 2021, 70(10): 10792-10803.
Dong T, Zhuang Z, Qi Q, et al. Intelligent Joint Network Slicing and Routing via GCN-powered Multi-Task Deep Reinforcement Learning[J]. IEEE Transactions on Cognitive Communications and Networking, 2021.
3. The names for Section 5.1, Algorithm 1 and Algorithm 2 can be improved, e.g., "Determine Resource Allocation" to "Resource Allocation Determination".
4. It is suggested to add markers when plotting Figures 8, 10, 11, 13, 14, 16, 19, 20, 22, 23, 25, and 26.
Reviewer 4 Report
Optimizing the allocation of the network and computation resources across multiple network slices is a critical but extremely difficult problem. Considering this challenge the authors proposed two heuristic algorithms to perform dynamic path routing and resource allocation for multi-tenant network slices in a two-tier architecture. The authors elaborately discussed the contributions, related works. Also presented comparative study of the existing works and their problems. The authors have done extensive simulation works to show that both algorithms significantly outperform the UFLOP algorithm proposed in previous work. The reference are very recent. Seems the authors are aware of the recent research. The presentation is very details. The authors need to address following minor comments.
A. Analyze the complexity of the algorithms
B. Summarize the Shortcoming the shortcomings of the previous works and how the proposed algorithms addressed the issue.
C. What is the limitation of the proposed work.
English is fine. Please re-visit the writings to fix minor spelling errors
